# Deep Reinforcement Learning Ensemble for Detecting Anomaly in Telemetry Water Level Data

Thakolpat Khampuengson [1,2,*] and Wenjia Wang [1]

1   School of Computing Sciences, University of East Anglia, Norwich NR4 7TJ, UK
2   Hydro-Informatics Institutes, Bangkok 10900, Thailand
*   Correspondence: t.khampuengson@uea.ac.uk

**Abstract:** Water levels in rivers are measured by various devices installed mostly in remote locations along the rivers, and the collected data are then transmitted via telemetry systems to a data centre for further analysis and utilisation, including producing early warnings for risk situations. So, the data quality is essential. However, the devices in the telemetry station may malfunction and cause errors in the data, which can result in false alarms or missed true alarms. Finding these errors requires experienced humans with specialised knowledge, which is very time-consuming and also inconsistent. Thus, there is a need to develop an automated approach. In this paper, we firstly investigated the applicability of Deep Reinforcement Learning (DRL). The testing results show that whilst they are more accurate than some other machine learning models, particularly in identifying unknown anomalies, they lacked consistency. Therefore, we proposed an ensemble approach that combines DRL models to improve consistency and also accuracy. Compared with other models, including Multilayer Perceptrons (MLP) and Long Short-Term Memory (LSTM), our ensemble models are not only more accurate in most cases, but more importantly, more reliable.

**Keywords:** anomaly detection; deep reinforcement learning; telemetry water level; time series; ensemble

## 1. Introduction

As climate change becomes more apparent, strong storms that bring heavy rainfalls occur with unusual patterns in many parts of the world. They can cause severe floods that result in devastating damages to infrastructure and loss of human life. In Thailand, flooding occurs more frequently and can cause enormous damages and huge economic losses of up to $46.5 billions a year [1]. On the other hand, drought happened in several parts of Thailand in 2015, notably in the Chao Phraya River Basin, the largest river basin in Thailand. This is consistent with a report from the UNDRR (2020) [2] that the ongoing drought crisis from 2015 to 2016 was the most severe drought in Thailand in 20 years. Therefore, it is essential to monitor water levels around the country because they form an important basis for making decisions on early warning.

In order to monitor the water levels in rivers, the Hydro Informatics Institute (HII) has been studying, building, and deploying water level telemetry stations around Thailand since 2013. Every ten minutes, each station transmits the measured data to the HII data centre through cellular or satellite networks. However, the water level data collected from telemetry station sensors might be incorrect due to some factors, such as human or animal activity, malfunctioning equipment, or interference of items surrounding the sensors. Any irregularity in the data might result in an inaccurate decision, such as false alarms or missed true alarms. Although water level data may be manually reviewed before being distributed for further analysis, the procedure necessitates the use of skilled specialists who examine the data from each station and make judgments about any probable abnormalities that may exist. This process is slow, very time-consuming and also unreliable. This motivates us to

develop an automated approach that can identify irregularities in a more accurate, efficient, and reliable manner.

In our previous work [3], we studied seven statistics-based models for detecting the anomalies. We found that although an individual model can be used to identify anomalies, it produces too many false alarms for some situations, such as when the water level will dramatically rise before a flood occurs, which is a scenario notably different from the others, and hence led to that the majority of statistical models identify such points as anomalous. We also created two ensembles as the ensemble methods [4], if constructed properly, have been demonstrated to be able to improve accuracy and reliability over individual models. The first ensemble was built with a simple strategy as it just combines some selected models with majority voting as its decision-making function. However, the test results showed that the simple ensemble models did not work well enough, even though they were usually better than most of the basic individual models. We then developed a complex ensemble method. It basically builds an ensemble of some simple ensembles selected from the candidates with some criteria, and these simple ensembles' outputs are combined with a weighted function. The findings indicate that a complex ensemble can improve the accuracy and consistency in recognising both abnormal and normal data.

In recent decades, deep machine learning methods have been demonstrated to be more powerful than conventional machine learning techniques in tackling complex problems such as speech recognition, handwriting recognition, image recognition, and natural language processing. One of these methods is the Long Short-Term Memory (LSTM) [5], outperformed the Multilayer Perceptron (MLP), although trained with only normal data, for detecting anomaly patterns from ECG signals. Moreover, the C-LSTM methods, which integrated a convolutional neural network (CNN), well performed to detect anomaly signals that are difficult to classify in web traffic data as shown in [6]. Another deep neural network based on anomaly detection technique was recently proposed, called DeepAnt, which consists of a time series predictor that uses CNN to predict the values of the next time step and classify the predicted values as normal or abnormal by passing them to the anomaly detector [7].

Reinforcement Learning (RL) is an algorithm that imitates the human learning process. It is based on the self-learning process in which an agent learns by interacting with the environment without any assumptions or rules. With the advantage of being able to learn on their own, it can identify unknown anomalies [8], which gives it an edge over other models. RL has been applied to a variety of applications such as games [9,10], robotics [11,12], natural language processing [13,14], computer vision [15], etc. It has also been used in some studies to detect anomalies in data, such as an experiment [16] that shows the use of the deep Q-function network (DQN) algorithm to detect anomalies in time series. Network intrusion detection systems (NIDS) are developed by [17], based on deep reinforcement learning. They utilised it to identify anomalous traffic on the campus network with a combination of flexible switching, learning, and detection modes. When the detection model performs below the threshold, the model is retrained. In the comparison against three traditional machine learning approaches, their model outperformed on two benchmark datasets, NSL-KDD and UNSW-NB15. A binary imbalanced classification model based on deep reinforcement learning (DRL) was introduced in [18]. They developed the reward function by setting the rewards for the minority class to be greater than the rewards for the majority class, which made DRL paying more attention to the minority class. They compared it to seven imbalanced learning methods and found that it outperformed other models in text datasets and extremely imbalanced data sets.

Although deep learning and RL methods have achieved excellent results in time series, one common issue is that their performance varies and it is hard to predict when they do better and when they perform relatively poor. In order to improve their consistency and accuracy, ensemble methods can be used. One example of such a method is the technique called particle swarm optimization (PSO), which was developed [19] to predict the changing trend of the Mexican Stock Exchange by combining several neural networks.

An ensemble of MLP, Backpropagation network (BPN), and LSTM, as shown in [20], was used to make models for detecting anomalous traffic in a network. The ensemble approach that utilises DRL schemes to maximise investment in stock trading was developed in [21]. They trained a DRL agent and obtained an ensemble trading strategy using three different actor-critics-based algorithms that outperformed the individual algorithm and two baselines in terms of the risk-adjusted return. Another ensemble RL that employed three types of deep neural networks in Q-learning and used ensemble techniques to make the final decision to increase prediction accuracy for wind speed short-term forecasting was suggested [22].

We discovered that none of the DRL methods have been applied to identify anomalies in telemetry water level data. We wonder whether DRL is applicable for identifying abnormalities in telemetry water level data. Even if the final DRL models perform well on training data, there is no guarantee that they will also perform well on testing data. Previous research has shown that combining many models that were trained in different ways may be more accurate than any of the individual models. So, in this paper, we aim to answer the following two research questions.

($Q_1$) Is DRL applicable and effective for identifying abnormalities in water level data?

($Q_2$) Can we build some ensembles of DRL to improve accuracy and consistency?

To answer them, in this paper, we conducted intensive investigation by evaluated the accuracy of DRL models with real-world data. Then we proposed a strategy to build some ensembles by selecting some suitable DRL models. The testing results show that DRL is applicable for identifying abnormalities in telemetry water level data with the advantage of identifying an unknown anomaly. However, the process of training takes a long time. The constructed ensembles not only improve accuracy and consistency, but also reduce the rate of false alarms.

Thus, the main contributions of this paper are:

($C_1$) DRL models have been demonstrated to be able to detect anomalies in telemetry water level data.

($C_2$) The ensembles we have constructed in this research with some suitable DRL models and use a weighted decision-making strategy can improve both accuracy and consistency. The proposed approach has a potential to be further developed and implemented for real-world application.

The rest of the paper is organised as follows: Section 2 overviews related work for anomaly detection. Section 3 describes the methodology. Section 4 presents the experiment design-from data preparation, parameters configurations, to evaluation metrics. Results and discussions are provided in Sections 5 and 6; the conclusion and suggestions for further work are summarised in Section 7.

## 2. Related Work

There are many methods for detecting anomalies in time series data. One basic approach is to use statistics-based methods, as reviewed in [23,24]. For example, simple and exponential smoothing techniques were used to identify anomalies in a continuous data stream of temperature in an industrial steam turbine [25]. But in general whilst they provided a baseline, they have a disadvantage in handling trends and periotics, e.g., the water level will dramatically rise before the flood, which differs considerably from the other data points and may lead to an increased false alarm rate. In addition, they can be affected by the types of anomaly and some work well for a certain type of problem. For example, for missing and outlier values, when the data is normally distributed, the *K*-means clustering method [26] is usually used, as it is simple and relatively effective. However, there is unfortunately no general guideline for choosing a method for a given problem.

Change Point Detection (CPD) is an important method for time series analysis. It indicates an unexpected and significant change in the analysed time series stream data and has been studied in many fields, as surveyed in [27,28]. However, the CPD has no ability to detect anomalies since not all detected change points are abnormalities. Many studies are being conducted to solve this problem by integrating CPD with other models to increase anomaly detection effectiveness. For example, researchers from [29] presented new techniques, called rule-based decision systems, that combine the results of anomaly detection algorithms with CPD algorithms to produce a confidence score for determining whether or not a data item is indeed anomalous. They tested their suggested method using multivariate water consumption data collected from smart metres, and the findings demonstrated that anomaly detection can be improved. Moreover, it has been proposed to detect anomalies in file transfer by using the CPD to detect the current bandwidth status from the server, then using this to calculate the expected file transfer time. The server administrator has been notified when observed file transfers take longer than expected, which may mean it may have something wrong [30]. The author of [31] investigated the CUSUM algorithm for change point detection to detect SYN flood attacks. The results demonstrated that the proposed algorithm provided robust performance with both high and low intensity attacks. Although change point detection performed well in many domains, the majority of them focused on changes in the behaviour of time series data (sequence anomaly) rather than point anomaly, which is my primary research emphasis. Furthermore, water level data at certain stations is strongly periodic with tidal effects, resulting in numerous data points changing from high tides to low tides each day, which is typical behaviour.

In recent decades, machine learning methods, including deep neural networks (DNNs), have been satisfactorily implemented in various hydrological issues such as outlier detection [32,33], water level prediction [34,35], data imputation [36], flood forecasting [37], streamflow estimation [38], etc. For example, in [39], the authors proposed the R-ANFIS (GL) method for modelling multistep-ahead flood forecasts of the Three Gorges Reservoir (TGR) in China, which was developed by combining the recurrent adaptive-network-based fuzzy inference system (R-ANFIS) with the genetic algorithm and the least square estimator (GL). The authors of [40] presented a flood prediction by comparing the expected typhoon tracking and the historical trajectory of typhoons in Taiwan in order to predict hydrographs from rainfall projections impacted by typhoons. The PCA-SOM-NARX approach was developed by [41] to forecast urban floods, combining the advantages of three models. Principal component analysis was used to derive the geographical distributions of urban floods (PCA). To construct a topological feature map, high-dimensional inundation recordings were grouped using a self-organizing map (SOM). To build 10-minute-ahead, multistep flood prediction models, nonlinear autoregressive with exogenous inputs (NARX) was utilised. The results showed that not only did the PCA-SOM-NARX approach produce more stable and accurate multistep-ahead flood inundation depth forecasts, but it was also more indicative of the geographical distribution of inundation caused by heavy rain events. Even though we can use forecasting methods to find anomalies by using prediction error as a threshold to classify data points as normal or not, it may take time to find the suitable threshold for each station.

An autoencoder is an unsupervised learning neural network. It is comprised of two parts: an *encoder* and a *decoder*. The encoder uses the concepts of dimension reduction algorithms to convert the original data into the different representations with the underlying structure of the data remaining and ignoring the noise. Meanwhile, the decoder reconstructs the data from the output of the encoder with as close of a resemblance as possible to the original data. An autoencoder is effectively used to solve many applied problems, from face recognition [42,43] and anomaly detection [44–47] to noise reduction [48–50]. In the time series domain, the authors of [51] proposed two autoencoder ensemble frameworks for unsupervised outlier identification in time series data based on sparsely connected recurrent neural networks, which addressed the issues from [52] given

the poor results when using an autoencoder with time series data. In one of the frameworks called the Independent Framework, multiple autoencoders are trained independently of one another, whereas in the other framework, the Shared Framework, multiple autoencoders are trained jointly in a manner that is multitask learning. They experimented by using univariate and multivariate real-world datasets. Experimental results revealed that the suggested autoencoder ensembles with a shared framework outperform baselines and state-of-the-art approaches. However, a disadvantage of this method is its high memory consumption when training many autoencoders together. In the hydrological domain, the authors of [53] presented the SAE-RNN model, which combined the stacked autoencoder (SAE) with a recurrent neural network (RNN) for multistep-ahead flood inundation forecasting. They started with SAE to encode the high dimensionality of input datasets (flood inundation depths), then utilised an LSTM-based RNN model to predict multistep-ahead flood characteristics based on regional rainfall patterns, and then decoded the output by SAE into regional flood inundation depths. They conducted experiments on datasets of flood inundation depths gathered in Yilan County, Taiwan, and the findings demonstrated that SAE-RNN can reliably estimate regional inundation depths in practical applications.

Time series based on ensemble methods have recently attracted attention. In a study by [54], they introduced the method EN-RTON2, which is an ensemble model with real-time updating using online learning and a submodel for real-time water level forecasts. However, they experimented with fewer datasets, a smaller number of records, and lower data frequency than our datasets. Furthermore, the authors offered no indication of the time necessary for training models and forecasting, which may be inadequate in our case given the number of stations and frequency of data transmission. The ensemble models were proposed by [55], which applied the sliding window based ensemble method to find the anomaly pattern in sensor data for preventing machine failure. They used a combination of classical clustering algorithms and the principle of biclustering to construct clusters representing different types of structure. Then they used these structures in a one-class classifier to detect outliers. The accuracy of these methods was tested on a time series of real-world datasets from the production of industry. The results have verified the accuracy and the validity of the proposed methods.

Despite the fact that numerous studies have used different anomaly detection techniques to tackle problems in many domains, only a few have focused on finding anomalies in water level data. Furthermore, the various employed sensors, installation area, frequency of data transmission, and measurement purposes lead to a variety of types of anomalies. As a result, techniques that perform well with one set of data may not work well with another.

## 3. Materials and Methods

This section describes firstly how deep reinforcement learning is constructed for detecting anomalies in water level telemetry data; and then how an ensemble can be built effectively by selecting suitable individual models to improve the accuracy of anomaly detection. The frameworks of these investigations were implemented with Python and their code can be accessed via GitHub (https://github.com/khaitao/RL-Anomaly-Detection-Water-Level, The last check on 5 August 2022).

### 3.1. Reinforcement Learning (RL)

Reinforcement learning (RL) is a branch of machine learning and it is one of the most active areas of research in artificial intelligence (AI), which is growing rapidly with a wide variety of algorithms. It is goal-oriented learning. The learner, or agent, learns from the result, or rewards, of its actions without being taught what actions to take. The way in which the agent decides which action to perform depends on the policy, which can be in the form of a lookup table or a complex search process. So, a policy function defines the agent's behaviour in an environment.

Most techniques that are used to find the optimal policy for resolving the RL problem are based on the Markov decision process (MDP), whereby the probability of next state $s'$ depends only on the current state $s$ and action $a$. It is represented by five important variables [56]:

- A finite set of states ($S$), which may be discrete or continuous.
- A finite set of actions ($A$). The agent takes an action $a$ from the action set $A$, $a \in A$.
- A transition probability ($T(s, a, s')$), which is the probability to get from state $s$ to another state $s'$ with action $a$.
- A reward probability ($R(s, a, s') \in \mathbb{R}$), which is the reward after going from state $s$ to another state $s'$ with action $a$.
- A discount factor ($\gamma$), which focuses on controls the important immediate and future rewards and lies within 0 to 1, $\gamma \in [0, 1]$.

The goal of learning is to maximise the expected cumulative reward in each episode. The agent should try to maximise the reward from any state $s$. The total reward $R$ at state $s$ as the sum of current rewards and the total discounted reward at the next state $s'$, which can be represented as follow:

$$R(s) = R(s, a, s') + \gamma R(s')$$

The algorithm that has been widely used in RL is Q-learning. It tries to maximize the values from Q-function, as shown in Equation (1), which can be approximated using the Bellman equation, which represents how good it is for an agent to perform a particular action in a state $s$.

$$NewQ(s, a) = Q(s, a) + \alpha(r + \gamma \max Q'(s', a') - Q(s, a)) \tag{1}$$

where $\alpha$ is the learning rate, and $\max Q'(s', a')$ is the highest Q value between possible actions from the new state $s'$.

### 3.1.1. Deep Q-Learning Network

Q-learning has a limitation: it does not perform well with many states and actions. Furthermore, going through all the actions in each state would be time-consuming. Therefore, the deep Q-learning network [57] (DQN) has been developed to solve those issues by using a neural network (NN). The Q-value is approximated by an NN with weights $w$, instead of finding the optimal Q-value through all possible state-action pairs, and errors are minimized through gradient descent. The overall process of DRL is depicted in Figure 1.

An agent usually does not know what action is best at the beginning of training. It may select the greatest action that is the best based on history (exploitation) or may explore new possibilities that may be better or worse (exploration). However, when should an agent "exploit" rather than "explore"? This remains a challenge since if the chosen action results in a faulty selection, an agent may get stuck in incorrect learning for a time. The epsilon-greedy algorithm is a simple way to balance exploration and exploitation. It does this by randomly choosing between exploration and exploitation and using the hyperparameter $\epsilon$ to switch between random action and Q-values, as shown in Equation (2). The normal procedure is to begin with $\epsilon = 1.0$ and gradually lower it to a small value, such as 0.01.

$$a = \begin{cases} \text{select a random action } a & \text{with probability } \epsilon \\ \\ argmax_a Q(s, a) & \text{otherwise} \end{cases} \tag{2}$$

Moreover, we make a transition from one state $s$ to the next state $s'$ by performing some action $a$ and receive a reward $r$ as $T(s, a, s')$. So, neural networks may overfit with correlated experience from those transitions. So, we saved the transition information in a buffer called *replay memory* and trained the DQN with a random transition in replay

memory instead of training with last transitions. It will reduce the correlated experience of learning each time, and then it will reduce the overfitting of the model.

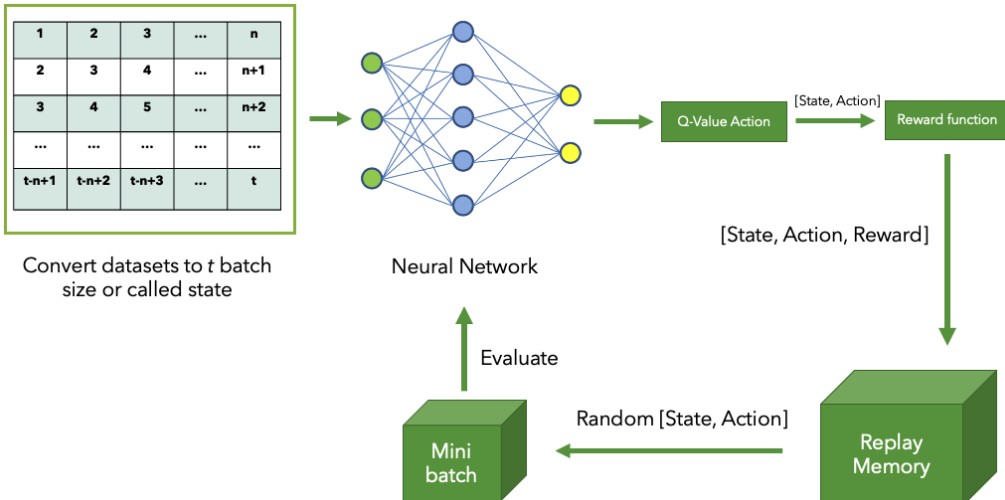

**Figure 1.** Overall process of DRL (Deep Reinforcement Learning).

3.1.2. Deep Reinforcement Learning Model (DRL)

The action of the DRL agent is to determine whether or not a data point is an abnormality. We assigned a value of 1 to the anomaly class and a value of 0 to the normal class. DQN was chosen as our reinforcement learning strategy. When state $s$ is received, an MLP is used as the RL agent's brain to generate Q-value, which is then followed by the $Q$ function. The epsilon decay approach is used for exploration and exploitation. In order to explore the entire environment space, we use the greedy factor $\epsilon$ to determine whether our DRL agent should follow the $Q$ function or randomly select an action.

For each iteration, DQN receives the set of states $S$ and predicts the label for training the DRL model. The transition is stored in replay memory. In each epoch, a mini batch of replay memory is sampled and used to train the model for loss minimization. Moreover, whether the model will learn well or not depends on the rewards function. The good reward function has an effect on the model's performance. If we offer a high reward for correctly identifying normal data in datasets, DRL may identify all data as normal in order to get the highest score. If, on the other hand, we give a high reward for finding outliers, DRL might label all data as outliers to get the best score.

Since our datasets are imbalanced, we will give the reward of the minority class higher than the majority class and give the penalty when our model misclassifies [18]. This will impact on the results in Q-values, then the model will select the best action to maximize the rewards. The reward function is defined below

$$rewards = \begin{cases} A & \text{predicted anomaly correct} \\ B & \text{predicted wrong} \\ C & \text{predicted normal correct} \end{cases} \quad (3)$$

A general issue in training neural networks is to determine how long they should be trained. Too few epochs may result in the model learning insufficiently, whereas too many epochs may result in the model overfitting. So, the performance of the model must be monitored during training by evaluating it on a validation data set at the end of each epoch and updating the model if the performance of the model on a validation is better than at the previous epoch. In our experiments, we selected 5 criteria as the conditions for generating the models: four performance metrics and the maximum number of epochs. The four measures are F1-score, the reward of each epoch, accuracy, and validation loss values. In the end, we will have five models: the finished training model ($DRL$), the models with

the highest F1-score ($DRL_{F1}$), the models with the highest rewards ($DRL_{Rwd}$), the model with the highest accuracy ($DRL_{Acc}$), and the model with the lowest validation loss values ($DRL_{Valid}$).

### 3.1.3. Ensemble Methods

In general, the capacity of an individual model is limited and may have only learned some parts of the problem, and hence may make mistakes in the areas where it has not learned sufficiently. Therefore, it can be useful to combine some individual models to form an ensemble to allow them to work collectively to compensate for each other's weaknesses. Many studies [4,58–60] have shown that if an ensemble is built with diverse models and appropriate decision-making functions, it can improve the accuracy of classification and also reliability. In our research, we created multiple ensembles by selecting suitable DRL models that had been generated from the previous experiments. We investigated two combining methods to aggregate the outputs from the member models of an ensemble: simple majority voting and weighted voting algorithms.

- *Majority Voting*: The predictions of each model in an ensemble have to be aggregated, and the final prediction is the class that gets the most votes. Each of our ensembles will be built with an odd number of classifiers in order to avoid a tie situation in voting.
- *Weighted Voting*: As the performance of individual models is usually different, treating them all the same way in decision marking appears unlogical, so we devised a weighted voting mechanism to take this difference into consideration when making a final decision in an ensemble. With the weighted voting method, the contribution from a model is weighed by its performance. For a model $m_i$, after it has been trained with the training data, its weight score $w_i$ is derived by using its $F_1$ score that is calculated on the given validation dataset; we then have a set of F1-scores of each model, $F_1m = \{F_1m_1, F_1m_2, ..., F_1m_M\}$. Then, these F1-scores are ranked to find the maximum and minimum scores. Finally, we calculate the normalised weighting score $w_i$ for module $m_i$ using the equation below:

$$w_i = \frac{F_1m_i - min(F_1m)}{max(F_1m) - min(F_1m)}, \; \forall \; i = 1, ..., M \tag{4}$$

The output of an ensemble, $\Phi(x)$, is calculated by multiplying the weight with the output of an individual module and taking the argument of maxima as follows:

$$\Phi(x) = argmax \sum_{i=1}^{M} w_i m_i(x) \tag{5}$$

where $M$ is the number of models in an ensemble, and $m_i(x)$ is the predicted class of model $i$.

### 3.2. Data Labelling

Water level data from telemetry stations were unlabelled for anomalies. It is then necessary to assign ground truth labels to all anomalies and normal data points in each time series of water level data in order to train the models with supervised algorithms. This was manually done by a group of the domain experts at the HII in a manner similar to the ensemble approach. Each specialist looked at the data and identified all the anomalies based on their experience. Then their judgements were aggregated by taking a consensus to decide if a data point is an anomaly or not.

### 3.3. Datasets

Since the DRL algorithm takes a lot of time for training on the computing facilities that we had, we were limited to consider some relatively small datasets. After data preprocessing, the 8 stations from the HII telemetry water level station were chosen for

use in this experiment, including CPY011, CPY012, CPY013, CPY014, CPY015, CPY016, CPY017, and YOM009. We chose the datasets from May and June for CPY011, CPY012, CPY013, CPY015, CPY016, and CPY017 in 2016 and similar months in 2015 for CPY014 and YOM009 because they have a low percentage of missing data. Figure 2 shows the water levels of these eight stations. It is visually clear that station YOM009 has very different behaviour from the others because it is located in a different region.

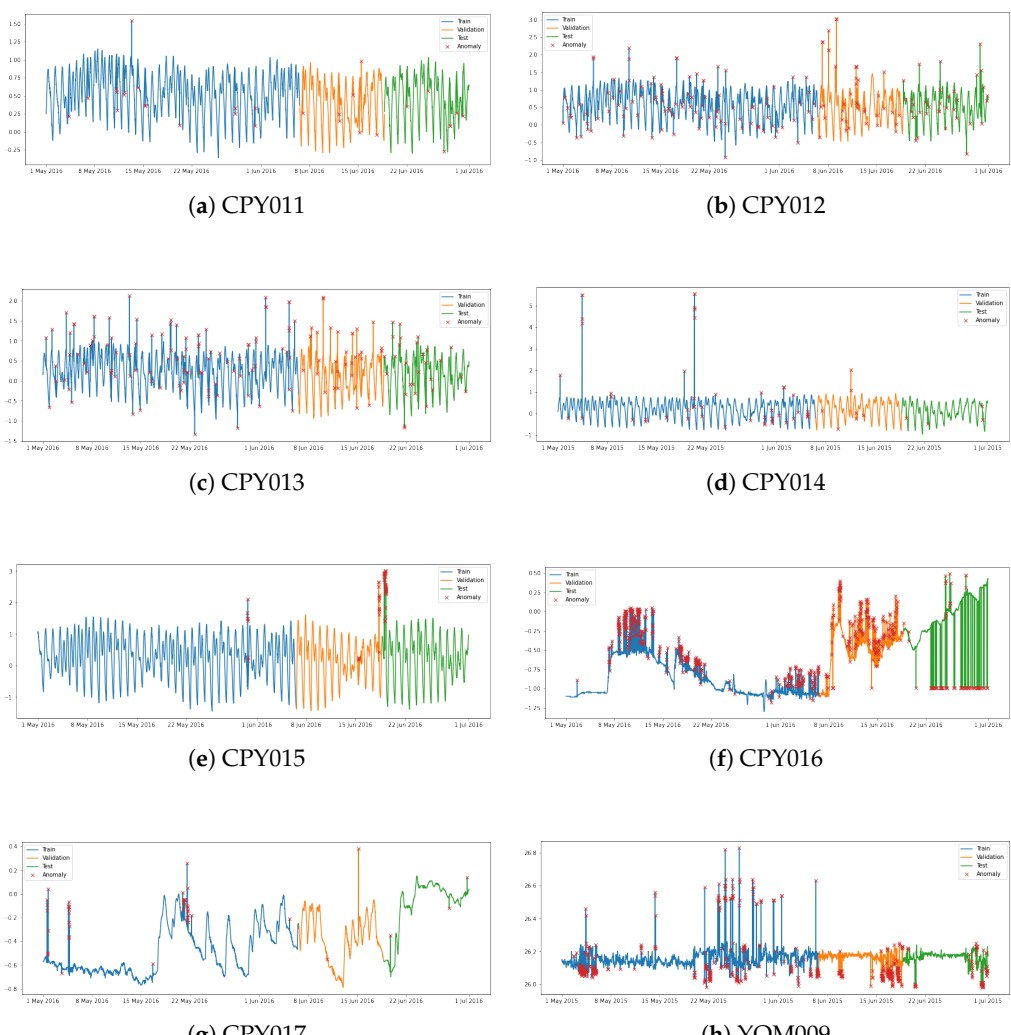

**Figure 2.** Water level data from eight stations: CPY011, CPY012, CPY013, CPY014, CPY015, CPY016, CPY017, and YOM009 (**a**–**h**). The different colours show the partitions of the data for training (blue), validation (orange) and testing (green). The anomalies are indicated by red crosses, x.

All the data are normalised and divided into 3 subsets, with the first 60% of a time series for training, the next 20% for validating, and the last 20% for testing, respectively. Table 1 shows the demographics of one partition of the data from each station. As can be seen, in general, the rates of anomalies are quite low for most stations, but the variances are considerably large. For example, they varied from 0.14% to 7.22% in the training data.

**Table 1.** Demographic summary of the water level data of 8 stations used in this research.

| Code | Training | | Validating | | Testing | | Total | |
|---|---|---|---|---|---|---|---|---|
| | Rec. | Anomaly | Rec. | Anomaly(%) | Rec. | Anomaly(%) | Rec. | Anomaly(%) |
| CPY011 | 5142 | 16(0.31%) | 1713 | 7(0.41%) | 1714 | 7(0.41%) | 8569 | 30(0.37%) |
| CPY012 | 5142 | 101(1.96%) | 1713 | 41(2.39%) | 1714 | 34(1.98%) | 8569 | 176(2.05%) |
| CPY013 | 5142 | 97(1.89%) | 1713 | 33(1.93%) | 1714 | 28(1.63%) | 8569 | 158(1.84%) |
| CPY014 | 5142 | 49(0.95%) | 1713 | 7(0.41%) | 1714 | 4(0.23%) | 8569 | 60(0.70%) |
| CPY015 | 5142 | 7(0.14%) | 1713 | 15(0.88%) | 1714 | 34(1.98%) | 8569 | 56(0.65%) |
| CPY016 | 5142 | 367(7.14%) | 1713 | 220(12.84%) | 1714 | 107(6.24%) | 8569 | 694(8.10%) |
| CPY017 | 5142 | 42(0.82%) | 1713 | 2(0.12%) | 1714 | 3(0.18%) | 8569 | 47(0.55%) |
| YOM009 | 5142 | 417(7.22%) | 1713 | 173(10.97%) | 1714 | 81(3.79%) | 8569 | 624(7.28%) |
| Avg. | 5142 | 137(2.66%) | 1713 | 62(3.63%) | 1714 | 37(2.17%) | 8569 | 231(2.69%) |

*3.4. Evaluation Metrics and Comparison Methods*

As our task is basically a classification problem. We therefore chose some commonly used measures: *Recall, Precision*, and *F1*, to evaluate the accuracy of models. They are defined by the following equations, based on the confusion matrix shown in Table 2.

**Table 2.** Confusion matrix of classification results.

| Actual/Predicted | Anomaly | Normal |
|---|---|---|
| Anomaly | TP | FN |
| Normal | FP | TN |

$$Recall = \frac{TP}{TP + FN}$$

$$Precision = \frac{TP}{TP + FP}$$

$$F1 = 2\frac{Precision * Recall}{Precision + Recall}$$

where $TP$, $FP$, $FN$, and $TN$ denote the number of true positive—correct predictions for anomaly data, false positive—the number of incorrect predictions for anomaly data, false negative—the number of incorrect predictions for normal data, and true negative—the number of correct predictions for normal data, respectively.

To make statistical comparisons, we implemented a statistically rigorous test for multiple classifiers across many datasets. This approach was initially described in [61] and is intended to examine the statistical significance of classifiers. This technique takes the strategy of testing the null hypothesis against the alternative hypothesis. The null hypothesis states that no difference exists between the average rankings of $k$ algorithms on $N$ datasets. The alternative hypothesis is that at least one algorithm's average rank differs.

In the first place, the $k$ methods are ranked according to their performance over the $N$ datasets; then, the average ranking of each algorithm is calculated. To test the null hypothesis, the Friedman test is calculated using Equation (6).

$$\chi_F^2 = \frac{12N}{k(k+1)} \left[ \sum_j R_j^2 - \frac{k(k+1)^2}{4} \right] \tag{6}$$

where $R_j$ is the rank of the $j$th of $k$ algorithms on $N$ datasets and the statistic is estimate using a chi-squared distribution with $k - 1$ degrees of freedom.

If the null hypothesis is rejected at the selected significance level $\alpha$, the post-hoc Nemenyi test is used to compare all classifiers to each other. The Nemenyi test is similar

to the Tukey test for ANOVA and uses a critical difference (CD), which is presented in Equation (7)

$$CD = q_\alpha \sqrt{\frac{k(k=1)}{6N}} \tag{7}$$

where $q_\alpha$ is calculated by the difference in the range of standard deviations between the smallest valued sample and the largest valued sample. The results of these tests are often visualised using a critical difference (CD) diagram. Classifiers are shown on a number line based on their average rank across all datasets, and bold CD lines are used to connect classifiers that are not significantly different.

In comparison, the performance of our approach, MLP, and LSTM have been used with the same number of hidden layers and the number of neurons in each hidden layer.

## 4. Experiment Design and Setting

### 4.1. Four Sets of Experiments

We designed four sets of experiments to test DRL models and ensemble models. (1) to train various DRL models and test them with the different data sampled from the same water level monitoring stations; (2) to train various DRL models with the data from a station and then test them with the data from other stations; (3) to build several ensembles by selecting different numbers of the DRL models and test them with the testing data from the same stations; and (4) to test the ensembles with the data from different stations. The purpose of doing these cross-station testing is to check and evaluate the generalisation ability of the DLR models and the ensembles.

### 4.2. Parameter Setting

For the DRL model, a multilayer perceptron network was used in the Q-network with the following parameters: the number of input nodes in the input layer was 36, one hidden layer with 18 nodes, and 2 nodes in the output layer. Moreover, epsilon-greedy policy ($\epsilon$) was used for exploration from 0.1 to 0.0001. The size of replay memory is 50,000, discount factor of intermediate rewards $\gamma$ was 0.99. The Adam algorithm was used to optimise the parameters of Q-Network and the learning rate was 0.001. The batch size was 256, training with 100, 500, 1000, 5000, and 10,000 episodes. The episode was over when the number of incorrectly identified anomalies was greater than the number of certain anomalies in the training set or had been trained on all the samples in the training set. We set the reward function parameters for $A$, $B$, and $C$ to be 0.9, $-0.1$, and 0.1, respectively. Furthermore, the window size of 6 was chosen to save time during the training process.

For comparison, MLP and LSTM were used with the identical structures as we used in DRL. They were trained using 100 epochs with early stopping to avoid overfitting. For each setting, the experiments were repeated 10 times with variations, and then the means and standard deviations of the results are reported in the next section.

### 4.3. Computing Facilities

All the experiments were coded with Python Programming Language (V3.6) (Python Software Foundation, https://www.python.org/, accessed on 30 June 2022) and TensorFlow 2.8, and run on a personal computer with an Intel Core i5-7500 CPU @ 3.4 GHz, 32 GB RAM, 64-Bit Operating System.

## 5. Results

### 5.1. Accuracies of DRL Models

For each station, various DRL models were generated over a range of epochs from 100 to 10,000, with the intention of investigating how well our proposed DRL method learns at the different points of training. The results are shown in Table 3.

Using the CPY011 dataset, we observed that $DRL$ and $DRL_{Rwd}$ with 1000 training iterations not only earned the highest F1-score of 0.8333, 0.7143 recall, and 1.0000 precision but also provided the highest average F1-score of 0.7433. However, after 1000 epochs of

training, the performance of all models, with the exception of $DRL_{Valid}$ decreased and then rose when 10,000 epochs were used.

**Table 3.** The performance of DRL when increasing the learning epochs (the best F1-score of each row shown in bold).

| Station | Epochs | DRL | | | $DRL_{F1}$ | | | $DRL_{Rwd}$ | | | $DRL_{Acc}$ | | | $DRL_{Valid}$ | | |
|---|---|---|---|---|---|---|---|---|---|---|---|---|---|---|---|---|
| | | Recall | Prec | F1 | Recall | Prec | F1 | Recall | Prec | F1 | Recall | Prec | F1 | Recall | Prec | F1 |
| CPY011 | 100 | 0.8571 | 0.5455 | 0.6667 | 0.8571 | 0.7500 | **0.8000** | 0.8571 | 0.5455 | 0.6667 | 0.8571 | 0.7500 | **0.8000** | 0.7143 | 0.6250 | 0.6667 |
| | 500 | 0.8571 | 0.7500 | **0.8000** | 0.8572 | 0.5455 | 0.6667 | 0.8571 | 0.7500 | **0.8000** | 0.8571 | 0.5455 | 0.6667 | 0.8571 | 0.6667 | 0.7500 |
| | 1000 | 0.7143 | 1.0000 | **0.8333** | 0.8571 | 0.6667 | 0.7500 | 0.7143 | 1.0000 | **0.8333** | 0.8571 | 0.6667 | 0.7500 | 0.4286 | 0.2727 | 0.3333 |
| | 5000 | 0.7143 | 0.6250 | 0.6667 | 0.8571 | 0.5000 | 0.6316 | 0.7143 | 0.6250 | 0.6667 | 0.8571 | 0.5000 | 0.6316 | 0.7143 | 0.7143 | **0.7143** |
| | 10,000 | 0.8571 | 0.6667 | **0.7500** | 1.0000 | 0.5833 | 0.7368 | 0.8571 | 0.6667 | **0.7500** | 1.0000 | 0.5833 | 0.7368 | 0.8571 | 0.4000 | 0.5455 |
| | Avg | 0.8000 | 0.7174 | **0.7433** | 0.8857 | 0.6091 | 0.7170 | 0.8000 | 0.7174 | **0.7433** | 0.8857 | 0.6091 | 0.7170 | 0.7143 | 0.5357 | 0.6020 |
| | Std | 0.0782 | 0.1743 | 0.0760 | 0.0639 | 0.0997 | 0.0674 | 0.0782 | 0.1743 | 0.0760 | 0.0639 | 0.0997 | 0.0674 | 0.1749 | 0.1901 | 0.1689 |
| CPY012 | 100 | 0.7059 | 0.4000 | 0.5106 | 0.7647 | 0.7027 | **0.7324** | 0.6764 | 0.3898 | 0.4946 | 0.7059 | 0.6857 | 0.6957 | 0.7647 | 0.7027 | **0.7324** |
| | 500 | 0.7647 | 0.6341 | 0.6933 | 0.7941 | 0.7297 | **0.7606** | 0.7353 | 0.6250 | 0.6757 | 0.7941 | 0.7297 | **0.7606** | 0.7941 | 0.7297 | **0.7606** |
| | 1000 | 0.6765 | 0.6571 | 0.6667 | 0.7647 | 0.6667 | 0.7123 | 0.6176 | 0.6000 | 0.6087 | 0.7647 | 0.6667 | 0.7123 | 0.7647 | 0.4062 | 0.5306 |
| | 5000 | 0.7059 | 0.7059 | 0.7059 | 0.6176 | 0.6176 | 0.6176 | 0.7059 | 0.7273 | **0.7164** | 0.6765 | 0.6970 | 0.6866 | 0.7059 | 0.7273 | **0.7164** |
| | 10,000 | 0.6471 | 0.7586 | 0.6984 | 0.7059 | 0.8000 | 0.7500 | 0.7059 | 0.7742 | 0.7385 | 0.7059 | 0.8276 | 0.7619 | 0.7941 | 0.7714 | **0.7826** |
| | Avg | 0.7000 | 0.6311 | 0.6550 | 0.7294 | 0.7033 | 0.7146 | 0.6882 | 0.6233 | 0.6468 | 0.7294 | 0.7213 | **0.7234** | 0.7647 | 0.6675 | 0.7045 |
| | Std | 0.0436 | 0.1378 | 0.0821 | 0.0702 | 0.0684 | 0.0572 | 0.0446 | 0.1489 | 0.0984 | 0.0483 | 0.0637 | 0.0357 | 0.0360 | 0.1481 | 0.1005 |
| CPY013 | 100 | 0.8710 | 0.3506 | 0.5000 | 0.8710 | 0.6136 | **0.7200** | 0.9032 | 0.3836 | 0.5385 | 0.8710 | 0.6136 | **0.7200** | 0.6774 | 0.1214 | 0.2059 |
| | 500 | 0.6774 | 0.3684 | 0.4773 | 0.8065 | 0.5000 | 0.6173 | 0.7419 | 0.3966 | 0.5169 | 0.8065 | 0.5000 | 0.6173 | 0.8387 | 0.4906 | **0.6190** |
| | 1000 | 0.8065 | 0.5952 | 0.6849 | 0.8065 | 0.5682 | 0.6667 | 0.7097 | 0.6111 | 0.6567 | 0.8065 | 0.5682 | 0.6667 | 0.9677 | 0.5556 | **0.7059** |
| | 5000 | 0.7742 | 0.5714 | 0.6575 | 0.6774 | 0.6774 | **0.6774** | 0.8065 | 0.5682 | 0.6667 | 0.6774 | 0.6364 | 0.6562 | 0.6774 | 0.5250 | 0.5915 |
| | 10,000 | 0.7097 | 0.6667 | 0.6875 | 0.8387 | 0.7647 | **0.8000** | 0.7742 | 0.6857 | 0.7273 | 0.8387 | 0.7647 | **0.8000** | 0.7419 | 0.6571 | 0.6970 |
| | Avg | 0.7678 | 0.5105 | 0.6014 | 0.8000 | 0.6248 | **0.6963** | 0.7871 | 0.5290 | 0.6212 | 0.8000 | 0.6166 | 0.6920 | 0.7806 | 0.4699 | 0.5639 |
| | Std | 0.0770 | 0.1423 | 0.1039 | 0.0736 | 0.1015 | 0.0685 | 0.0743 | 0.1337 | 0.0899 | 0.0736 | 0.0978 | 0.0706 | 0.1237 | 0.2045 | 0.2061 |
| CPY014 | 100 | 0.7500 | 0.7500 | **0.7500** | 0.7500 | 0.7500 | 0.7500 | 0.7500 | 0.7500 | **0.7500** | 0.7500 | 0.7500 | **0.7500** | 0.7500 | 0.7500 | **0.7500** |
| | 500 | 0.7500 | 0.3750 | 0.5000 | 0.7500 | 0.7500 | 0.7500 | 0.7500 | 1.0000 | **0.8571** | 0.7500 | 0.7500 | 0.7500 | 0.7500 | 1.0000 | **0.8571** |
| | 1000 | 0.7500 | 0.3750 | 0.5000 | 0.7500 | 0.5000 | 0.6000 | 0.7500 | 0.3750 | 0.5000 | 0.7500 | 0.5000 | 0.6000 | 0.7500 | 0.5000 | 0.6000 |
| | 5000 | 0.7500 | 0.3333 | 0.4615 | 0.7500 | 0.5000 | 0.6000 | 0.7500 | 0.3333 | 0.4615 | 0.7500 | 0.5000 | 0.6000 | 0.7500 | 0.4286 | 0.5455 |
| | 10,000 | 0.2500 | 0.1667 | 0.2000 | 0.7500 | 0.6000 | **0.6667** | 0.2500 | 0.1667 | 0.2000 | 0.7500 | 0.6000 | **0.6667** | 0.7500 | 0.4286 | 0.5455 |
| | Avg | 0.6500 | 0.4000 | 0.4823 | 0.7500 | 0.6200 | **0.6733** | 0.6500 | 0.5250 | 0.5537 | 0.7500 | 0.6200 | **0.6733** | 0.7500 | 0.6214 | 0.6596 |
| | Std | 0.2236 | 0.2137 | 0.1952 | 0.0000 | 0.1255 | 0.0751 | 0.2236 | 0.3405 | 0.2584 | 0.0000 | 0.1255 | 0.0751 | 0.0000 | 0.2495 | 0.1385 |
| CPY015 | 100 | 0.2353 | 0.4706 | 0.3137 | 0.3235 | 0.5238 | 0.4000 | 0.2353 | 0.4706 | 0.3137 | 0.3235 | 0.5238 | **0.4000** | 0.1765 | 0.3529 | 0.2353 |
| | 500 | 0.2059 | 0.4667 | 0.2857 | 0.3235 | 0.5000 | **0.3929** | 0.2059 | 0.4667 | 0.2857 | 0.3235 | 0.5000 | **0.3929** | 0.1471 | 0.3846 | 0.2128 |
| | 1000 | 0.3824 | 0.4483 | 0.4127 | 0.3824 | 0.5000 | **0.4333** | 0.3824 | 0.4483 | 0.4127 | 0.3824 | 0.5000 | **0.4333** | 0.4118 | 0.4516 | 0.4308 |
| | 5000 | 0.2353 | 0.4706 | 0.3137 | 0.3235 | 0.5238 | **0.4000** | 0.2353 | 0.4706 | 0.3137 | 0.3235 | 0.5238 | **0.4000** | 0.2353 | 0.4706 | 0.3137 |
| | 10,000 | 0.3824 | 0.4483 | 0.4127 | 0.3824 | 0.5200 | **0.4407** | 0.3824 | 0.4483 | 0.4127 | 0.3824 | 0.5200 | **0.4407** | 0.3824 | 0.4483 | 0.4127 |
| | Avg | 0.2883 | 0.4609 | 0.3477 | 0.3471 | 0.5135 | **0.4134** | 0.2883 | 0.4609 | 0.3477 | 0.3471 | 0.5135 | **0.4134** | 0.2706 | 0.4216 | 0.3211 |
| | Std | 0.0868 | 0.0116 | 0.0604 | 0.0323 | 0.0124 | 0.0219 | 0.0868 | 0.0116 | 0.0604 | 0.0323 | 0.0124 | 0.0219 | 0.1202 | 0.0503 | 0.0995 |
| CPY016 | 100 | 0.6636 | 0.3100 | 0.4226 | 0.5981 | 0.5203 | **0.5565** | 0.6916 | 0.2960 | 0.4146 | 0.5981 | 0.5203 | **0.5565** | 0.6168 | 0.4889 | 0.5455 |
| | 500 | 0.6636 | 0.2763 | 0.3901 | 0.6355 | 0.4048 | 0.4945 | 0.6449 | 0.2727 | 0.3833 | 0.5981 | 0.5161 | **0.5541** | 0.5047 | 0.5094 | 0.5070 |
| | 1000 | 0.6355 | 0.2547 | 0.3636 | 0.5981 | 0.5333 | 0.5639 | 0.6449 | 0.2644 | 0.3750 | 0.6168 | 0.5641 | **0.5893** | 0.6168 | 0.4342 | 0.5097 |
| | 5000 | 0.5888 | 0.2727 | 0.3728 | 0.5888 | 0.6238 | **0.6058** | 0.3084 | 0.2089 | 0.2491 | 0.5421 | 0.6105 | 0.5743 | 0.5234 | 0.2902 | 0.3733 |
| | 10,000 | 0.5794 | 0.2366 | 0.3360 | 0.6168 | 0.4177 | 0.4981 | 0.6355 | 0.2208 | 0.3277 | 0.6262 | 0.5447 | **0.5826** | 0.6168 | 0.4177 | 0.4981 |
| | Avg | 0.6262 | 0.2701 | 0.3770 | 0.6075 | 0.5000 | 0.5438 | 0.5851 | 0.2526 | 0.3499 | 0.5963 | 0.5511 | **0.5714** | 0.5757 | 0.4281 | 0.4867 |
| | Std | 0.0402 | 0.0274 | 0.0321 | 0.0187 | 0.0904 | 0.0472 | 0.1562 | 0.0366 | 0.0644 | 0.0326 | 0.0384 | 0.0156 | 0.0567 | 0.0858 | 0.0659 |
| CPY017 | 100 | 1.0000 | 0.7500 | **0.8571** | 1.0000 | 0.5000 | 0.6667 | 1.0000 | 0.7500 | **0.8571** | 1.0000 | 0.7500 | **0.8571** | 1.0000 | 0.7500 | **0.8571** |
| | 500 | 1.0000 | 0.7500 | **0.8571** | 1.0000 | 0.5000 | 0.6667 | 1.0000 | 0.7500 | **0.8571** | 1.0000 | 0.3750 | 0.5455 | 0.6667 | 0.5000 | 0.5714 |
| | 1000 | 1.0000 | 0.2143 | 0.3529 | 1.0000 | 0.7500 | **0.8571** | 1.0000 | 0.2000 | 0.3333 | 1.0000 | 0.7500 | **0.8571** | 0.0000 | 0.0000 | - |
| | 5000 | 1.0000 | 0.5000 | **0.6667** | 1.0000 | 0.4286 | 0.6000 | 1.0000 | 0.5000 | **0.6667** | 1.0000 | 0.4286 | 0.6000 | 1.0000 | 0.3333 | 0.5000 |
| | 10,000 | 0.6667 | 0.6667 | 0.6667 | 0.6667 | 0.2857 | 0.4000 | 0.6667 | 0.6667 | 0.6667 | 1.0000 | 0.6000 | **0.7500** | 0.6667 | 0.6667 | 0.6667 |
| | Avg | 0.9333 | 0.5762 | 0.6801 | 0.9333 | 0.4929 | 0.6381 | 0.9333 | 0.5733 | 0.6762 | 1.0000 | 0.5807 | **0.7219** | 0.6667 | 0.4500 | 0.6488 |
| | Std | 0.1491 | 0.2266 | 0.2062 | 0.1491 | 0.1683 | 0.1641 | 0.1491 | 0.2323 | 0.2140 | 0.0000 | 0.1755 | 0.1443 | 0.4082 | 0.2982 | 0.1547 |
| YOM009 | 100 | 0.6308 | 0.3178 | 0.4227 | 0.5692 | 0.3033 | 0.3957 | 0.6462 | 0.3182 | **0.4264** | 0.5692 | 0.3033 | 0.3957 | 0.5692 | 0.3394 | 0.4253 |
| | 500 | 0.5846 | 0.2734 | 0.3725 | 0.6769 | 0.3121 | 0.4272 | 0.6923 | 0.3020 | 0.4206 | 0.4769 | 0.4769 | **0.4769** | 0.4769 | 0.4769 | **0.4769** |
| | 1000 | 0.5385 | 0.2966 | 0.3825 | 0.6769 | 0.2973 | 0.4131 | 0.5538 | 0.3103 | 0.3978 | 0.4615 | 0.3947 | **0.4255** | 0.5846 | 0.3016 | 0.3979 |
| | 5000 | 0.5538 | 0.1818 | 0.2738 | 0.6615 | 0.3644 | **0.4699** | 0.6154 | 0.2581 | 0.3636 | 0.4923 | 0.4103 | 0.4476 | 0.5077 | 0.4177 | 0.4583 |
| | 10,000 | 0.4769 | 0.2627 | 0.3388 | 0.5692 | 0.2741 | 0.3700 | 0.4769 | 0.2605 | 0.3370 | 0.5385 | 0.2917 | 0.3784 | 0.4308 | 0.4308 | **0.4308** |
| | Avg | 0.5569 | 0.2665 | 0.3581 | 0.6307 | 0.3102 | 0.4152 | 0.5969 | 0.2898 | 0.3891 | 0.5077 | 0.3754 | 0.4248 | 0.5138 | 0.3933 | **0.4378** |
| | Std | 0.0570 | 0.0519 | 0.0558 | 0.0565 | 0.0334 | 0.0373 | 0.0839 | 0.0285 | 0.0382 | 0.0449 | 0.0776 | 0.0395 | s0.0640 | 0.0712 | 0.0306 |

The top models to identify anomalies on the CPY012 dataset are $DRL_{Valid}$, with a maximum F1-score of 0.7826 after 10,000 training epochs. However, $DRL_{Acc}$ obtained the greatest average F1-score with 0.7234. Meanwhile, 10,000 training epochs with $DRL_{F1}$ and $DRL_{Acc}$ delivered the highest F1-score for identifying anomalies in CPY013 data, at 0.8000 F1-score. Furthermore, $DRL_{F1}$ provided the highest average F1-score of 0.6963.

With just 500 epochs of training on CPY014 data, $DRL_{Rwd}$ and $DRL_{Valid}$ delivered the best F1-score of 0.8571. However, the maximum average F1-score achieved by $DRL_{F1}$ and $DRL_{Acc}$ was just 0.6733. When looking at the results on CPY015 data, the best models are $DRL_{F1}$ and $DRL_{Acc}$. This is shown by the fact that their F1-scores were the highest in many training epochs.

$DRL_{Acc}$ was the best model for detecting anomalies in CPY016 data since it not only had the greatest F1-score in almost every training epoch but also had the highest average F1-score of 0.5714. Meanwhile, every model scored the best F1-score of 0.8571, 100 percent recall, and 0.7500 accuracy when trained with 100 epochs on CPY017, with the exception of the $DRL_{F1}$ model, which achieved just 0.6667 F1-score. While the best models for detecting

anomalies on YOM009 are $DRL_{Acc}$ and $DRL_{Valid}$, which both have the same F1-score of 0.4769, the worst models are $DRL$ while training with 5000 iterations at a 0.2728 F1-score, 0.5538 recall, and 0.1818 precision.

Figure 3 shows the comparison of the critical differences between the different DRL models. The number associated with each algorithm is the average rank of the DRL models on each type of dataset, and solid bars represent groups of classifiers with no significant difference. There is no statistically significant difference across the models, with $DRL_{Acc}$ ranking first, followed by $DRL_{F1}$, $DRL$, $DRL_{Rwd}$, and $DRL_{Valid}$ ranking last.

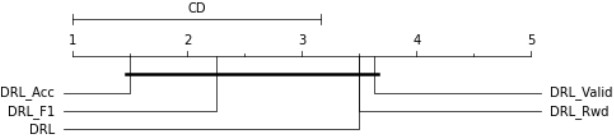

**Figure 3.** A critical difference diagram for 5 different DRL models on different datasets of telemetry water level data.

Figure 4 also shows a line graph of the F1-score as the number of epochs of training from each model increases. We can observe that as the number of epochs is increased, the performance of all deep reinforcement learning models using data from CPY012, CPY013, and CPY015 tends to improve. When training with CPY014 data, on the other hand, the F1-score of each model tends to stay the same or go down as the number of epochs goes up. In the case of trained models with CPY016 data, the F1-score of each model tends to stabilise and slightly decrease, with the exception of $DRL_{Valid}$, which tends to grow after 5000 epochs of training. When we looked at the models that were trained with the CPY017 dataset, the F1-score of $DRL_{F1}$ went up after training with 1000 epochs and then went down. Other models, however, went up when training with more epochs, even though the performance of some models went down after 1000 epochs, while the F1-score of models that have been trained with CPY011 and YOM009 remained stable when training with more epochs.

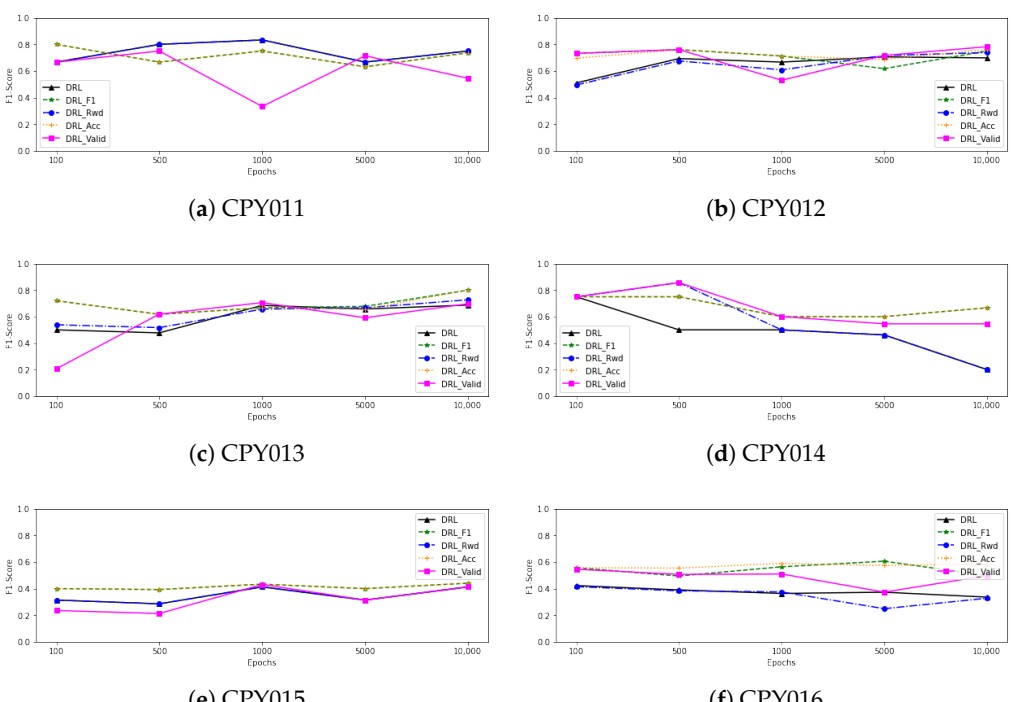

(**a**) CPY011

(**b**) CPY012

(**c**) CPY013

(**d**) CPY014

(**e**) CPY015

(**f**) CPY016

**Figure 4.** *Cont.*

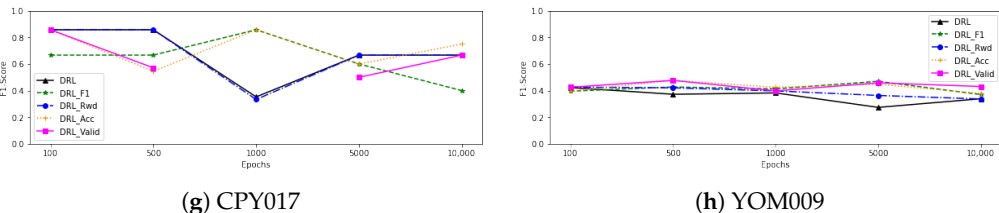

**(g)** CPY017             **(h)** YOM009

**Figure 4.** F1-score when increasing the learning epochs at each station.

Figure 5 shows the findings of the best DRL model for each station. We can observe that the DRL model performs well, capturing the majority of abnormalities in testing datasets. However, it still did not work well when there were anomalies in data that changed frequently, like when there were anomalies in YOM009 data between 29 June and 1 July 2015, and in CPY015 data on 19 June 2016.

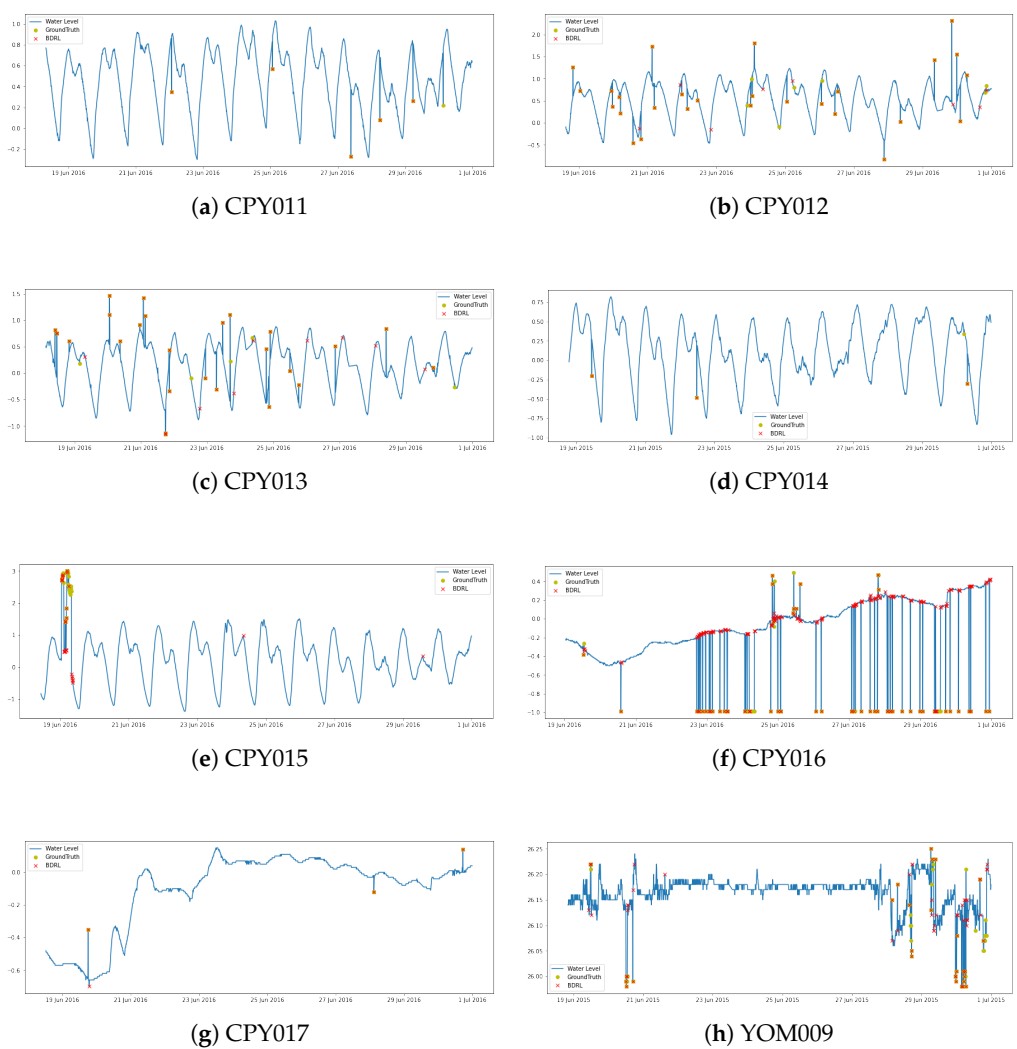

**Figure 5.** Anomaly detection from the best DRL model of each station. (**a**) CPY011 with $DRL$; (**b**) CPY012 with $DRL_{Valid}$; (**c**) CPY013 with $DRL_{F1}$; (**d**) CPY014 with $DRL_{Rwd}$; (**e**) CPY015 with $DRL_{F1}$; (**f**) CPY016 with $DRL_{F1}$; (**g**) CPY017 with $DRL$; (**h**) YOM009 with $DRL_{Acc}$.

*5.2. Performance on the Same Station*

We evaluated the performance of our techniques with MLP and LSTM models on eight telemetry water level datasets. The data in each station is first divided into training, validating, and testing parts in a 6:2:2 ratio. The results were averaged after being run ten

times and then were compared to the averaged DRL models of each station as shown in Table 4. It demonstrated that $DRL_{F1}$ and $DRL_{Acc}$ had the highest average F1-scores for detecting anomalies on CPY015, with F1-scores of 0.4133. MLP had the greatest average F1-score when it came to detecting anomalies on CPY011, CPY012, and CPY014 with scores of 0.8505, 0.7822, and 0.8571, respectively. On the other stations, LSTM was the top performing model. According to the CD diagram in Figure 6, the best LSTM model had the greatest ranking of performance, followed by $DRL_{Acc}$ and MLP.

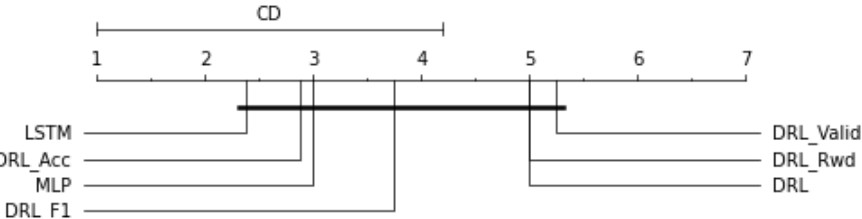

**Figure 6.** A critical difference diagram of each model.

**Table 4.** The mean F1-scores and standard deviation of all DRL, MLP, and LSTM models when testing with the dataset from different stations (the best F1-score of each row is shown in bold).

| Station | DRL | $DRL_{F1}$ | $DRL_{Rwd}$ | $DRL_{Acc}$ | $DRL_{Valid}$ | MLP | LSTM |
|---|---|---|---|---|---|---|---|
| CPY011 | 0.7433 (±0.08) | 0.7170 (±0.07) | 0.7433 (±0.08) | 0.7170 (±0.07) | 0.6020 (±0.17) | **0.8505 (±0.06)** | 0.8167 (±0.04) |
| CPY012 | 0.6550 (±0.08) | 0.7146 (±0.06) | 0.6468 (±0.10) | 0.7234 (±0.04) | 0.7045 (±0.10) | **0.7822 (±0.03)** | 0.7753 (±0.02) |
| CPY013 | 0.6014 (±0.10) | 0.6963 (±0.07) | 0.6212 (±0.09) | 0.6920 (±0.07) | 0.5639 (±0.21) | 0.6998 (±0.03) | **0.7265 (±0.02)** |
| CPY014 | 0.4823 (±0.20) | 0.6733 (±0.08) | 0.5537 (±0.26) | 0.6733 (±0.08) | 0.6596 (±0.14) | **0.8571 (±0.00)** | **0.8571 (±0.00)** |
| CPY015 | 0.3477 (±0.06) | **0.4134 (±0.02)** | 0.3477 (±0.06) | **0.4134 (±0.02)** | 0.3211 (±0.10) | 0.2220 (±0.10) | 0.3276 (±0.09) |
| CPY016 | 0.3770 (±0.03) | 0.5438 (±0.05) | 0.3499 (±0.06) | 0.5714 (±0.02) | 0.4867 (±0.07) | 0.5651 (±0.14) | **0.6252 (±0.06)** |
| CPY017 | 0.6801 (±0.21) | 0.6381 (±0.16) | 0.6762 (±0.21) | 0.7219 (±0.14) | 0.6488 (±0.15) | 0.9778 (±0.07) | **0.9857 (±0.05)** |
| YOM009 | 0.3581 (±0.06) | 0.4152 (±0.04) | 0.3891 (±0.04) | 0.4248 (±0.04) | **0.4378 (±0.03)** | 0.2358 (±0.05) | 0.2596 (±0.06) |

We discovered that $DRL_{F1}$ and $DRL_{Acc}$ had the highest average F1-scores for detecting anomalies on CPY015, with F1-scores of 0.4133. MLP had the greatest average F1-score when it came to detecting anomalies on CPY011, CPY012, and CPY014 with scores of 0.8505, 0.7822, and 0.8571, respectively. On the other stations, LSTM was the top performing model. The LSTM model has the highest ranking of performance, according to the CD diagram in Figure 6, followed by $DRL_{Acc}$ and MLP.

Since RL models need time to learn until they have enough knowledge to do their task, time costing is the one important thing that we need to be interested in. We calculate the time spent by the best deep learning models (BDRL) and comparative models, as shown in Table 5. The MLP model requires the least training time per epoch, with an average of 0.30 s, followed by the LSTM model at 0.64 s, and the DRL model at 17.56 s. For MLP and LSTM training with early stopping, they needed an average of 12 and 15 training epochs, respectively, while our method requires around 4638 epochs to get optimal results. It means that the MLP model took an average of 2.97 s to train, while LSTM took 9.20 s and DRL took an average of 78,756 s, which is about 22 h.

**Table 5.** The number of training epochs and the time spent on each epoch for each model.

| Station | Training Epochs | | | Time (s./epochs) | | | Total Time (s.) | | |
|---|---|---|---|---|---|---|---|---|---|
| | MLP | LSTM | BDRL | MLP | LSTM | BDRL | MLP | LSTM | BDRL |
| CPY011 | 12 | 17 | 1000 | 0.17 | 0.64 | 18.66 | 2.04 | 10.88 | 18,666 |
| CPY012 | 15 | 19 | 10,000 | 0.16 | 0.62 | 18.20 | 2.40 | 11.78 | 182,000 |
| CPY013 | 11 | 17 | 10,000 | 0.18 | 0.64 | 17.88 | 1.98 | 10.88 | 178,000 |
| CPY014 | 11 | 6 | 500 | 0.55 | 0.83 | 18.32 | 6.05 | 4.98 | 2490 |
| CPY015 | 7 | 11 | 10,000 | 0.24 | 0.62 | 16.03 | 1.68 | 6.82 | 160,300 |
| CPY016 | 17 | 21 | 5000 | 0.17 | 0.51 | 16.58 | 2.89 | 10.71 | 82,900 |
| CPY017 | 13 | 17 | 100 | 0.20 | 0.56 | 16.74 | 2.6 | 9.52 | 1674 |
| YOM009 | 6 | 11 | 500 | 0.69 | 0.73 | 18.82 | 4.14 | 8.03 | 4015 |
| avg | 12 | 15 | 4638 | 0.30 | 0.64 | 17.65 | 2.97 | 9.20 | 78,756 |

*5.3. Performance on the Different Station*

After generating various models on some stations' data and testing them with the same stations, we tested these models with the data collected from different stations with the intention of examining their generalisation ability. The F1-scores of each model are provided in Table 6.

**Table 6.** The F1-scores of the best DRL models when testing with the dataset from same station (show in the bracket) and different stations, while the average F1-scores and standard deviations of each station were calculated without their own scores.

| Tested Dataset | Trained Dataset | | | | | | | |
|---|---|---|---|---|---|---|---|---|
| | CPY011 | CPY012 | CPY013 | CPY014 | CPY015 | CPY016 | CPY017 | YOM009 |
| CPY011 | (0.8333) | 0.1667 | 0.1967 | 0.0255 | 0.4138 | 0.0596 | 0.2000 | 0.1556 |
| CPY012 | 0.6000 | (0.7826) | 0.7164 | 0.1136 | 0.0533 | 0.5088 | 0.6667 | 0.6667 |
| CPY013 | 0.6531 | 0.6667 | (0.8000) | 0.1875 | 0.1034 | 0.5421 | 0.5970 | 0.5952 |
| CPY014 | 0.4000 | 0.8571 | 0.8571 | (0.8571) | 0.0000 | 0.2727 | 0.5871 | 0.6000 |
| CPY015 | 0.0000 | 0.1538 | 0.1474 | 0.0672 | (0.4407) | 0.0900 | 0.2078 | 0.0839 |
| CPY016 | 0.5369 | 0.6129 | 0.6550 | 0.1276 | 0.4103 | (0.6058) | 0.1203 | 0.6703 |
| CPY017 | 0.5000 | 0.3529 | 0.5455 | 0.2308 | 0.0000 | 0.1765 | (0.8571) | 0.4000 |
| YOM009 | 0.0000 | 0.3297 | 0.1905 | 0.0771 | 0.0000 | 0.4189 | 0.3158 | (0.4769) |
| avg | 0.3843 | 0.4485 | 0.4727 | 0.1185 | 0.1401 | 0.2955 | 0.3850 | **0.4531** |
| std | 0.2742 | 0.2680 | 0.2908 | 0.0713 | 0.1896 | 0.1975 | 0.2257 | 0.2457 |

Using $DRL_{Rwd}$-the best model for detecting anomalies by training with CPY011 data and then identifying anomalies from other stations, we can see that, though it works rather well, with F1-scores ranging from 0.4 on CPY014 to 0.65 on CPY013 data, it is unable to detect anomalies on CPY015 and YOM009. Using the BDRL model of the CPY012 training dataset, $DRL_{Valid}$, although it provided good performance when identifying anomalies in the CPY013, CPY014, and CPY016 datasets with F1-scores greater than 0.61, especially CPY014 with a 0.8571 f1-score, which more than detected anomalies on its own dataset, it provided poor performance, with an F1-score lower than 0.4000, when detecting anomalies in other stations. Similar to $DRL_{F1}$, which was trained using CPY013 data, it not only performs well when recognising anomalies on its own dataset but also when detecting anomalies on the CPY014 dataset, with an F1-score of 0.8571. The BDRL model, $DRL_{Rwd}$, that was trained with CPY014 did the worst when it was used to find anomalies in other stations' data, with an F1-score of less than 0.23 for every dataset and the lowest F1-score of only 0.0255 for CPY011. Similar to the best model on CPY015 datasets, which performed poorly, with the highest F1-score on CPY011 data being 0.4138 and being unable to identify anomalies on CPY014, CPY017, and YOM009. Meanwhile, the best model for detecting anomalies on CPY016 data performed the best for detecting anomalies on CPY013 with a 0.5421 F1-score. The model that was trained on CPY017 did the best of finding anomalies in data from CPY012, CPY013, and CPY014 with an F1-score greater than 0.58. While the

best model from the YOM009 training dataset achieved a low F1-score on CPY011, CPY015, and CPY017, 0.0839 is the lowest F1-score. However, when it was used to find outliers on COY012, CPY013, CPY014, and CPY016 with F1-scores higher than 0.59, it did better than its own training data.

It is worth noting that models trained using CPY014 and CPY015 data perform poorly when used to identify anomalies from other stations. This may be due to the fact that the actual number of anomalies in those stations are relatively low and most of them are kind of extreme outliers, as shown in Figure 2, so the models were trained with only those kinds of anomalies, which may not be enough for the model to learn. In contrast to YOM009, which has a many number and types of anomalies for model to learn, as a result, it can identify abnormalities on CPY012, CPY013, CPY014, and CPY016 better than other models that were trained with another station.

Then, we tested MLP and LSTM using data from different stations to compare our method to the candidate models. Table 7 represents the results of the MLP models when tested with the datasets from the same and different stations. Using the CPY011 dataset, the MLP models achieved the highest F1-score of 0.5430 on CPY016, despite their being unable to identify anomalies on CPY014 and YOM009. Similar to finding anomalies on CPY012, it offered good results with F1-scores of more than 0.63, with the exception of CPY011, CPY015, and YOOM009, which produced F1-scores of less than 0.4. The best MLP of the CPY013 training dataset provided the highest F1-score on the CPY014 dataset (0.8571 F1-score) and the lowest on CPY015 (0.2093 F1-score). Anomalies on the YOM009 dataset were the most difficult for the MLP models trained on CPY014 to detect, with an F1-score of just 0.1818. However, it performed excellent results in identifying anomalies on CPY017 with a 1.0000 F1-score. Meanwhile, the MLP model on the CPY015 dataset performed poorly when detecting abnormalities from other stations. On the other hand, the MLP models that were trained on CPY016 and CPY017 generated good results when used to identify anomalies from other stations, despite still performing poorly in some stations. In contrast, the MLP model trained on YOM009 worked well when used to detect abnormalities on other stations but performed badly when detecting anomalies on its own data. Furthermore, it performed well on CPY017 data, with a 1.000 F1-score.

**Table 7.** The F1-scores of the MLP models when testing with the dataset from the same station (shown in the bracket) and different stations.

| Tested Dataset | Trained Dataset | | | | | | | |
| | CPY011 | CPY012 | CPY013 | CPY014 | CPY015 | CPY016 | CPY017 | YOM009 |
|---|---|---|---|---|---|---|---|---|
| CPY011 | (0.9231) | 0.4000 | 0.2917 | 0.7778 | 0.0000 | 0.1373 | 0.7059 | 0.6087 |
| CPY012 | 0.4889 | (0.8387) | 0.8060 | 0.7500 | 0.0541 | 0.6923 | 0.7273 | 0.7857 |
| CPY013 | 0.4390 | 0.6786 | (0.7500) | 0.7000 | 0.0000 | 0.7123 | 0.6909 | 0.7719 |
| CPY014 | 0.0000 | 0.8571 | 0.8571 | (0.8571) | 0.0000 | 0.6000 | 0.8571 | 0.8571 |
| CPY015 | 0.1951 | 0.1509 | 0.2093 | 0.2593 | (0.3529) | 0.1420 | 0.3077 | 0.2414 |
| CPY016 | 0.5430 | 0.6380 | 0.6587 | 0.6322 | 0.0915 | (0.6629) | 0.5665 | 0.6550 |
| CPY017 | 0.5000 | 0.6667 | 0.6000 | 1.0000 | 0.0000 | 0.2857 | (1.0000) | 1.0000 |
| YOM009 | 0.0000 | 0.2308 | 0.2955 | 0.1818 | 0.0000 | 0.4404 | 0.1772 | (0.2857) |
| avg | 0.3094 | 0.5174 | 0.5312 | 0.6144 | 0.0208 | 0.4300 | 0.5761 | **0.7028** |
| std | 0.2396 | 0.2609 | 0.2643 | 0.2929 | 0.0371 | 0.2473 | 0.2460 | 0.2408 |

In the case of the LSTM model, as depicted in Table 8. They performed well, with an average F1-score of more than 0.42 for each station except CPY015, which had an average F1-score of 0.1099. However, they generated poor performances in some stations, such as the LSTM of CPY016 that achieved an F1-score of only 0.1754 when used to detect anomalies on the CPY011 dataset, and it was unable to detect anomalies on CPY014, CPY017, and YOM009 datasets with the LSTM that had been trained on the CPY015 dataset. However, it provided excellent performance when detecting anomalies on CPY017 with the LSTM that has been trained on the CPY014 dataset. When the LSTM was trained

on YOM009, it did well at finding anomalies from other stations, especially CPY014 and CPY017, with an F1-score of 0.8571.

**Table 8.** The F1-scores of the LSTM models when testing with the dataset from the same station (shown in the bracket) and different stations.

| Tested Dataset | Trained Dataset | | | | | | | |
|---|---|---|---|---|---|---|---|---|
| | CPY011 | CPY012 | CPY013 | CPY014 | CPY015 | CPY016 | CPY017 | YOM009 |
| CPY011 | (0.8571) | 0.4828 | 0.2333 | 0.5000 | 0.3077 | 0.1754 | 0.7059 | 0.5185 |
| CPY012 | 0.6400 | (0.8387) | 0.8308 | 0.7368 | 0.0444 | 0.7536 | 0.7500 | 0.7458 |
| CPY013 | 0.5652 | 0.7333 | (0.7463) | 0.7213 | 0.1081 | 0.6857 | 0.7333 | 0.7188 |
| CPY014 | 0.4000 | 0.8571 | 0.8571 | (0.8571) | 0.0000 | 0.8571 | 0.8571 | 0.8571 |
| CPY015 | 0.3043 | 0.3636 | 0.2340 | 0.3939 | (0.4364) | 0.2045 | 0.3704 | 0.2564 |
| CPY016 | 0.5679 | 0.6897 | 0.6824 | 0.5634 | 0.3089 | (0.6630) | 0.6296 | 0.6359 |
| CPY017 | 0.5000 | 1.0000 | 0.5455 | 1.0000 | 0.0000 | 0.3333 | (1.0000) | 0.8571 |
| YOM009 | 0.0000 | 0.2619 | 0.3146 | 0.2727 | 0.0000 | 0.4190 | 0.2857 | (0.3542) |
| avg | 0.4253 | 0.6269 | 0.5282 | 0.5983 | 0.1099 | 0.4898 | 0.6189 | **0.6557** |
| std | 0.2190 | 0.2679 | 0.2717 | 0.2430 | 0.1410 | 0.2747 | 0.2111 | 0.2129 |

Furthermore, we generated a bar chart to compare the average F1-score from each model when tested with the data collected from different stations, as shown in Figure 7. When evaluated with data from other stations, the models trained with CPY012 and CPY013 produced an average F1-score greater than 0.4. The models trained on CPY015 earned poor performance when used to identify anomalies from other stations, with an average F1-score lower than 0.2. DRL models that were trained with CPY015 outperform other models in detecting anomalies in data from other stations. LSTM models trained on CPY011, CPY012, CPY016, and CPY017, on the other hand, outperform other models in detecting abnormalities on other datasets. When trained with data from CPY013, CPY014, and YOM009, MLP had the best F1-score for finding outliers in other datasets.

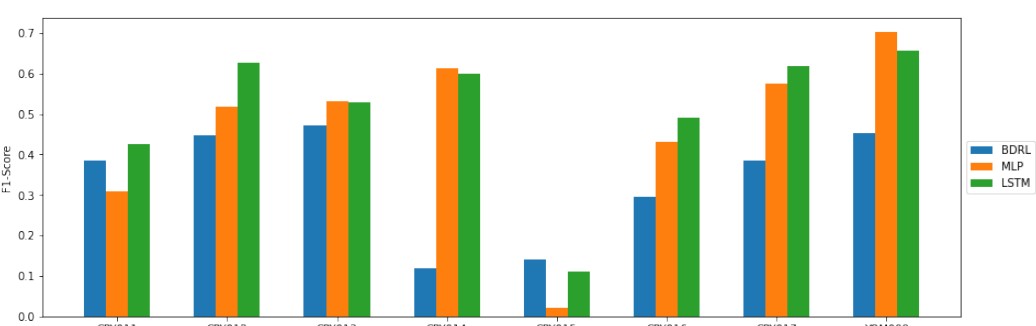

**Figure 7.** Bar charts of average F1-scores of the DRL, MLP, and LSTM when tested with the data collected from different stations.

### 5.4. Ensemble Results

Since we have multiple RL models after each epoch of training, and since each model performs the best in each of the criteria, we then built an ensemble that combined the decisions of all RL models, with the aim of generating a better final decision. In model selection, we select all five models and select the three models with the highest ranking in F1-score to build our ensemble model. For decision making, we used majority voting and weighted voting strategies to make a final decision. So, we have 4 ensemble models for each epoch of training, including a majority voting ensemble model with 3 ($EDRL_3$) and 5 ($EDRL_5$) models, and a weighted ensemble model with 3 ($WEDRL_3$) and 5 ($WEDRL_5$) models.

### 5.4.1. Performance on the Same Station

The results of our ensemble models are shown in Table 9 demonstrated that ensemble with majority voting and weighted voting that were generated from the top three *DRL* models of CPY011 provided the best with 0.8333 F1-score, while $WDRL_3$ that was generated from the DRL model after trained with 10,000 epochs is the best model to detect anomalies in CPY012 datasets with an F1-score of 0.7941. The ensemble model of CPY013 that performs the best is $EDRL_3$ and $WEDRL_3$ at 0.8000. The best ensemble model for identifying anomalies in CPY014 datasets is the ensemble model that provided the F1-score of 0.8571. With CPY015 data, the models with the highest F1-score are $EDRL_3$, $WEDRL_3$, and $WEDRL_5$. These models were built based on the individual DRL model, which was trained for 10,000 iterations. Meanwhile, $WEDRL_3$ got the highest F1-score of 0.5922 for CPY016 by combining the best three DRL models that were trained over 5000 iterations. With CPY017, $EDRL_5$ outperforms other ensemble models with a 100 percent in every metric. The ensemble results of YOM009, $WEDRL_5$, offered the highest performance with an F1-score of 0.5032 that was generated from the *DRL* model after 500 epochs of training.

**Table 9.** The performance of ensemble models (the best F1-score of each row is shown in bold).

| Station | Epochs | $EDRL_3$ | | | $EDRL_5$ | | | $WEDRL_3$ | | | $WEDRL_5$ | | |
|---|---|---|---|---|---|---|---|---|---|---|---|---|---|
| | | Recall | Prec | F1 | Recall | Prec | F1 | Recall | Prec | F1 | Recall | Prec | F1 |
| CPY011 | 100 | 0.8571 | 0.7500 | **0.8000** | 0.7143 | 0.6250 | 0.6667 | 0.8571 | 0.7500 | **0.8000** | 0.8571 | 0.7500 | **0.8000** |
| | 500 | 0.8571 | 0.7500 | **0.8000** | 0.8571 | 0.6667 | 0.7500 | 0.8571 | 0.7500 | **0.8000** | 0.8571 | 0.6667 | 0.7500 |
| | 1000 | 0.7143 | 1.0000 | **0.8333** | 0.7143 | 0.8333 | 0.7692 | 0.7143 | 1.0000 | **0.8333** | 0.8571 | 0.7500 | 0.8000 |
| | 5000 | 0.7143 | 0.6250 | 0.6667 | 0.7143 | 0.6250 | 0.6667 | 0.7143 | 0.7143 | **0.7143** | 0.7143 | 0.7143 | **0.7143** |
| | 10,000 | 0.8571 | 0.6667 | 0.7500 | 1.0000 | 0.7778 | **0.8750** | 0.8571 | 0.6667 | 0.7500 | 0.8571 | 0.6000 | 0.7059 |
| | Avg | 0.8000 | 0.7583 | 0.7700 | 0.8000 | 0.7056 | 0.7455 | 0.8000 | 0.7762 | **0.7795** | 0.8285 | 0.6962 | 0.7540 |
| | std | 0.0782 | 0.1455 | 0.0650 | 0.1278 | 0.0949 | 0.0863 | 0.0782 | 0.1297 | 0.0471 | 0.0639 | 0.0637 | 0.0451 |
| CPY012 | 100 | 0.7647 | 0.7027 | 0.7324 | 0.7353 | 0.7353 | **0.7353** | 0.7647 | 0.7027 | 0.7324 | 0.7647 | 0.6842 | 0.7222 |
| | 500 | 0.7941 | 0.7297 | **0.7606** | 0.7941 | 0.7297 | **0.7606** | 0.7941 | 0.7297 | **0.7606** | 0.7941 | 0.7105 | 0.7500 |
| | 1000 | 0.7647 | 0.6667 | 0.7123 | 0.7353 | 0.8621 | **0.7937** | 0.7647 | 0.6667 | 0.7123 | 0.7353 | 0.6410 | 0.6849 |
| | 5000 | 0.7059 | 0.7273 | 0.7164 | 0.7059 | 0.7500 | 0.7273 | 0.7059 | 0.7273 | 0.7164 | 0.7353 | 0.7353 | **0.7353** |
| | 10,000 | 0.7059 | 0.8276 | 0.7619 | 0.7059 | 0.8276 | 0.7619 | 0.7941 | 0.7941 | **0.7941** | 0.7353 | 0.8333 | 0.7812 |
| | Avg | 0.7471 | 0.7308 | 0.7367 | 0.7353 | 0.7809 | **0.7558** | 0.7647 | 0.7241 | 0.7432 | 0.7529 | 0.7209 | 0.7347 |
| | std | 0.0394 | 0.0598 | 0.0236 | 0.0360 | 0.0601 | 0.0261 | 0.0360 | 0.0466 | 0.0342 | 0.0263 | 0.0719 | 0.0355 |
| CPY013 | 100 | 0.8710 | 0.6136 | **0.7200** | 0.8387 | 0.6190 | 0.7123 | 0.8710 | 0.6136 | **0.7200** | 0.9032 | 0.5957 | 0.7179 |
| | 500 | 0.8065 | 0.5000 | 0.6173 | 0.7742 | 0.5714 | 0.6575 | 0.8387 | 0.4906 | 0.6190 | 0.8065 | 0.5556 | **0.6579** |
| | 1000 | 0.8065 | 0.6098 | 0.6944 | 0.8065 | 0.6098 | 0.6944 | 0.9355 | 0.6042 | **0.7342** | 0.9032 | 0.6087 | 0.7273 |
| | 5000 | 0.8065 | 0.5952 | 0.6849 | 0.7097 | 0.5789 | 0.6377 | 0.7097 | 0.6471 | 0.6769 | 0.7742 | 0.6486 | **0.7059** |
| | 10,000 | 0.8387 | 0.7647 | **0.8000** | 0.7419 | 0.6765 | 0.7077 | 0.8387 | 0.7647 | **0.8000** | 0.8387 | 0.7429 | 0.7879 |
| | Avg | 0.8258 | 0.6167 | 0.7033 | 0.7742 | 0.6111 | 0.6819 | 0.8387 | 0.6240 | 0.7100 | 0.8452 | 0.6303 | **0.7194** |
| | std | 0.0288 | 0.0949 | 0.0660 | 0.0510 | 0.0417 | 0.0328 | 0.0822 | 0.0983 | 0.0674 | 0.0577 | 0.0712 | 0.0467 |
| CPY014 | 100 | 0.7500 | 0.7500 | 0.7500 | 0.7500 | 1.0000 | **0.8571** | 0.7500 | 1.0000 | **0.8571** | 0.7500 | 0.7500 | 0.7500 |
| | 500 | 0.7500 | 1.0000 | **0.8571** | 0.7500 | 1.0000 | **0.8571** | 0.7500 | 1.0000 | **0.8571** | 0.7500 | 1.0000 | **0.8571** |
| | 1000 | 0.7500 | 0.5000 | 0.6000 | 0.7500 | 0.6000 | **0.6667** | 0.7500 | 0.3750 | 0.5000 | 0.7500 | 0.3750 | 0.5000 |
| | 5000 | 0.7500 | 0.5000 | **0.6000** | 0.7500 | 0.5000 | **0.6000** | 0.7500 | 0.5000 | **0.6000** | 0.7500 | 0.3750 | 0.5000 |
| | 10,000 | 0.7500 | 0.6000 | 0.6667 | 0.7500 | 0.6000 | 0.6667 | 0.7500 | 0.6000 | 0.6667 | 0.7500 | 0.7500 | **0.7500** |
| | Avg | 0.7500 | 0.6700 | 0.6948 | 0.7500 | 0.7400 | **0.7295** | 0.7500 | 0.6950 | 0.6962 | 0.7500 | 0.6500 | 0.6714 |
| | std | 0.0000 | 0.2110 | 0.1097 | 0.0000 | 0.2408 | 0.1196 | 0.0000 | 0.2896 | 0.1584 | 0.0000 | 0.2710 | 0.1625 |
| CPY015 | 100 | 0.3235 | 0.5238 | **0.4000** | 0.2647 | 0.5000 | 0.3462 | 0.3235 | 0.5238 | **0.4000** | 0.2647 | 0.4737 | 0.3396 |
| | 500 | 0.3235 | 0.5000 | **0.3929** | 0.2059 | 0.4667 | 0.2857 | 0.3235 | 0.5000 | **0.3929** | 0.2941 | 0.5263 | 0.3774 |
| | 1000 | 0.3824 | 0.5000 | 0.4333 | 0.4118 | 0.4828 | **0.4444** | 0.3824 | 0.5000 | 0.4333 | 0.3824 | 0.4643 | 0.4194 |
| | 5000 | 0.3235 | 0.5238 | **0.4000** | 0.2353 | 0.4706 | 0.3137 | 0.3235 | 0.5238 | **0.4000** | 0.3235 | 0.5238 | **0.4000** |
| | 10,000 | 0.3824 | 0.5200 | **0.4407** | 0.3824 | 0.4483 | 0.4127 | 0.3824 | 0.5200 | **0.4407** | 0.3824 | 0.5200 | **0.4407** |
| | Avg | 0.3471 | 0.5135 | **0.4134** | 0.3000 | 0.4737 | 0.3605 | 0.3471 | 0.5135 | **0.4134** | 0.3294 | 0.5016 | 0.3954 |
| | std | 0.0323 | 0.0124 | 0.0219 | 0.0916 | 0.0192 | 0.0666 | 0.0323 | 0.0124 | 0.0219 | 0.0526 | 0.0300 | 0.0390 |
| CPY016 | 100 | 0.5981 | 0.5203 | 0.5565 | 0.6075 | 0.4962 | 0.5462 | 0.5981 | 0.5203 | 0.5565 | 0.6168 | 0.5366 | **0.5739** |
| | 500 | 0.5981 | 0.5203 | 0.5565 | 0.6168 | 0.3952 | 0.4818 | 0.6075 | 0.5603 | **0.5830** | 0.5981 | 0.5333 | 0.5639 |
| | 1000 | 0.6168 | 0.5323 | 0.5714 | 0.6542 | 0.4636 | 0.5426 | 0.6168 | 0.5546 | **0.5841** | 0.6168 | 0.5455 | 0.5789 |
| | 5000 | 0.5701 | 0.6100 | 0.5894 | 0.5514 | 0.4275 | 0.4816 | 0.5701 | 0.6162 | **0.5922** | 0.5888 | 0.3987 | 0.4755 |
| | 10,000 | 0.6168 | 0.4177 | 0.4981 | 0.6262 | 0.4295 | 0.5095 | 0.6262 | 0.5447 | **0.5826** | 0.6449 | 0.5111 | 0.5702 |
| | Avg | 0.6000 | 0.5201 | 0.5544 | 0.6112 | 0.4424 | 0.5123 | 0.6037 | 0.5592 | **0.5797** | 0.6131 | 0.5050 | 0.5525 |
| | std | 0.0191 | 0.0684 | 0.0343 | 0.0377 | 0.0386 | 0.0314 | 0.0215 | 0.0353 | 0.0135 | 0.0215 | 0.0608 | 0.0434 |
| CPY017 | 100 | 1.0000 | 0.7500 | **0.8571** | 1.0000 | 0.7500 | **0.8571** | 1.0000 | 0.7500 | **0.8571** | 1.0000 | 0.7500 | **0.8571** |
| | 500 | 1.0000 | 0.7500 | 0.8571 | 1.0000 | 1.0000 | **1.0000** | 1.0000 | 0.7500 | 0.8571 | 1.0000 | 0.7500 | 0.8571 |
| | 1000 | 1.0000 | 0.7500 | **0.8571** | 1.0000 | 0.2308 | 0.3750 | 1.0000 | 0.7500 | **0.8571** | 1.0000 | 0.7500 | **0.8571** |
| | 5000 | 1.0000 | 0.5000 | **0.6667** | 1.0000 | 0.5000 | **0.6667** | 1.0000 | 0.5000 | **0.6667** | 1.0000 | 0.5000 | **0.6667** |
| | 10,000 | 0.6667 | 0.6667 | 0.6667 | 0.6667 | 0.6667 | 0.6667 | 1.0000 | 0.6000 | **0.7500** | 0.6667 | 0.6667 | 0.6667 |
| | Avg | 0.9333 | 0.6833 | 0.7809 | 0.9333 | 0.6295 | 0.7131 | 1.0000 | 0.6700 | **0.7976** | 0.9333 | 0.6833 | 0.7809 |
| | std | 0.1491 | 0.1087 | 0.1043 | 0.1491 | 0.2868 | 0.2354 | 0.0000 | 0.1151 | 0.0866 | 0.1491 | 0.1087 | 0.1043 |

**Table 9.** *Cont.*

| Station | Epochs | EDRL$_3$ | | | EDRL$_5$ | | | WEDRL$_3$ | | | WEDRL$_5$ | | |
|---|---|---|---|---|---|---|---|---|---|---|---|---|---|
| | | Recall | Prec | F1 | Recall | Prec | F1 | Recall | Prec | F1 | Recall | Prec | F1 |
| YOM009 | 100 | 0.6308 | 0.3254 | **0.4293** | 0.5846 | 0.3115 | 0.4064 | 0.6154 | 0.3226 | 0.4233 | 0.6462 | 0.3281 | 0.4352 |
| | 500 | 0.4769 | 0.4769 | 0.4769 | 0.6154 | 0.3960 | 0.4819 | 0.4769 | 0.4769 | 0.4769 | 0.6000 | 0.4333 | **0.5032** |
| | 1000 | 0.5538 | 0.3600 | 0.4364 | 0.5538 | 0.3186 | 0.4045 | 0.5231 | 0.3778 | **0.4387** | 0.4923 | 0.3299 | 0.3951 |
| | 5000 | 0.5538 | 0.4091 | 0.4706 | 0.5846 | 0.4086 | 0.4810 | 0.6308 | 0.3981 | **0.4881** | 0.5538 | 0.3830 | 0.4528 |
| | 10,000 | 0.5385 | 0.2991 | 0.3846 | 0.4923 | 0.3048 | 0.3765 | 0.4154 | 0.3971 | **0.4060** | 0.4615 | 0.3158 | 0.3750 |
| | Avg | 0.5508 | 0.3741 | 0.4396 | 0.5661 | 0.3479 | 0.4301 | 0.5323 | 0.3945 | **0.4466** | 0.5508 | 0.3580 | 0.4323 |
| | std | 0.0548 | 0.0707 | 0.0371 | 0.0467 | 0.0501 | 0.0484 | 0.0914 | 0.0554 | 0.0350 | 0.0757 | 0.0494 | 0.0503 |

Figure 8 depicts line charts that indicate the F1-score of each ensemble model that was trained using data from each station. It was clear from the results that the ensemble models not only delivered good performances and had a tendency to either improve or keep their F1-scores steady but also reduced the false alarms by increasing the precision scores. When we compared the results of each training epoch of the individual DRL model and the ensemble model, as shown in Tables 3 and 9, we discovered that ensemble models performed better than every single DRL model in many training epochs. In particular, $EDRL_5$ on the CPY017 with 500 training epochs generated an excellent score of 1.0000 in every metrics index, resulting from a 25% increase in accuracy and a 15% increase in F1-score. Meanwhile, $EDRL_5$ on the CPY011 with 10,000 training epochs improved the performance of the best individual model with an F1-score from 0.75 to 0.8750, reached 1.00 in terms of recall, and increased precision by 20%. By combining the DRL models trained on only 500 epochs, the ensemble model on YOM009 got the highest F1-score of 0.5032.

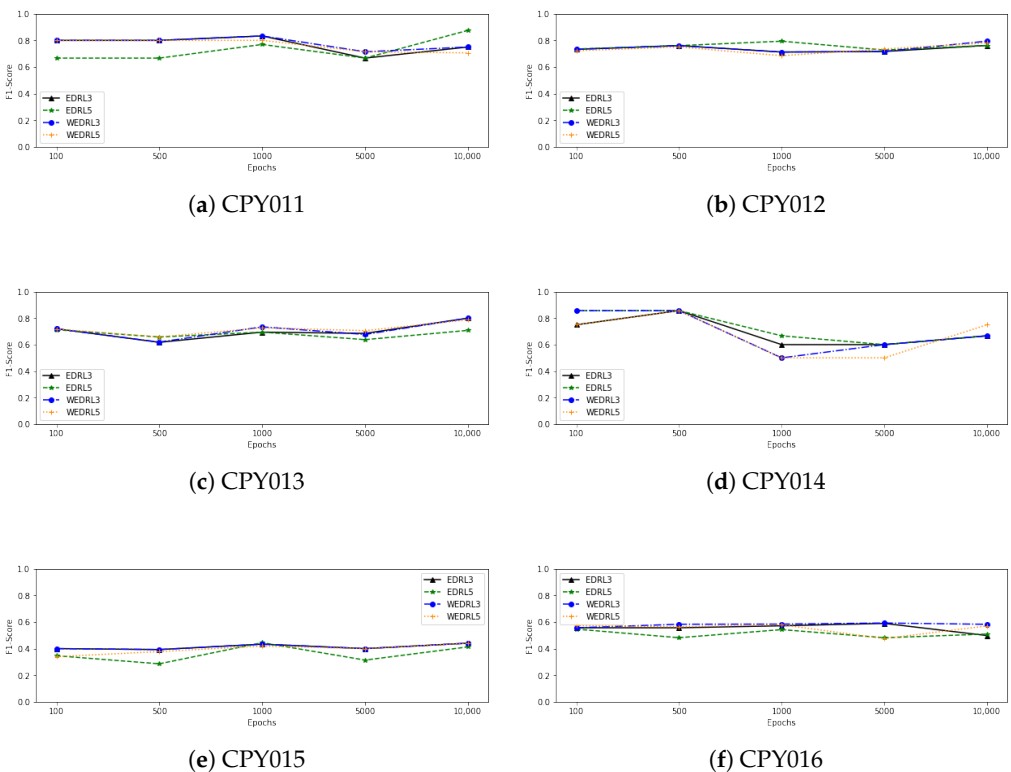

(**a**) CPY011

(**b**) CPY012

(**c**) CPY013

(**d**) CPY014

(**e**) CPY015

(**f**) CPY016

**Figure 8.** *Cont.*

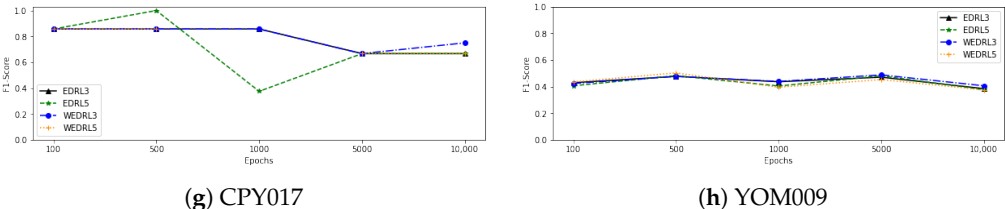

(**g**) CPY017            (**h**) YOM009

**Figure 8.** F1-score of ensemble model when increasing the learning epochs at CPY011, CPY012, CPY013, CPY014, CPY015, CPY016, CPY017, and YOM009 (**a**–**h**).

As shown in Table 10, we evaluated the average F1-score of each individual DRL model and ensemble of DRL models against the other neural network models. We can see that the LSTM model was the best model when detecting anomalies on CPY013, CPY014, CPY016, and CPY017, while $WEDRL_3$ provided the highest average F1-score on CPY015 and YOM009. The highest F1-score was 0.4134 on CPY015, which was provided by $DRL_{F1}$, $DRL_{Acc}$, $EDRL_3$, and $WEDRL_3$. Although MLP and LSTM beat other models in many datasets, $WEDRL_3$ has the greatest average ranking, as shown in Figure 9. In other words, the ensemble model not only has the potential to improve the performance of a single model, but it also has a higher reliability to deliver excellent performance than a single model.

**Table 10.** The mean F1-scores and standard deviations of all of the DRL, MLP, LSTM, and ensemble of DRL-based models when testing with the dataset from different stations (the best F1-score of each station is shown in bold).

| Models | CPY011 | CPY012 | CPY013 | CPY014 | CPY015 | CPY016 | CPY017 | YOM009 |
|---|---|---|---|---|---|---|---|---|
| *DRL* | 0.7433 (±0.08) | 0.6550 (±0.08) | 0.6014 (±0.10) | 0.4823 (±0.20) | 0.3477 (±0.06) | 0.3770 (±0.03) | 0.6801 (±0.21) | 0.3581 (±0.06) |
| $DRL_{F1}$ | 0.7170 (±0.07) | 0.7146 (±0.06) | 0.6963 (±0.07) | 0.6733 (±0.08) | **0.4134 (±0.02)** | 0.5438 (±0.05) | 0.6381 (±0.16) | 0.4152 (±0.04) |
| *DRLRwd* | 0.7433 (±0.08) | 0.6468 (±0.10) | 0.6212 (±0.09) | 0.5537 (±0.26) | 0.3477 (±0.06) | 0.3499 (±0.06) | 0.6762 (±0.21) | 0.3891 (±0.04) |
| $DRL_{Acc}$ | 0.7170 (±0.07) | 0.7234 (±0.04) | 0.6920 (±0.07) | 0.6733 (±0.08) | **0.4134 (±0.02)** | 0.5714 (±0.02) | 0.7219 (±0.14) | 0.4248 (±0.04) |
| $DRL_{Valid}$ | 0.6020 (±0.17) | 0.7045 (±0.10) | 0.5639 (±0.21) | 0.6596 (±0.14) | 0.3211 (±0.10) | 0.4867 (±0.07) | 0.6488 (±0.15) | 0.4378 (±0.03) |
| $EDRL_3$ | 0.7700 (±0.06) | 0.7367 (±0.02) | 0.7033 (±0.07) | 0.6948 (±0.11) | **0.4134 (±0.02)** | 0.5544 (±0.03) | 0.7809 (±0.10) | 0.4396 (±0.04) |
| $EDRL_5$ | 0.7455 (±0.09) | 0.7558 (±0.03) | 0.6819 (±0.03) | 0.7295 (±0.12) | 0.3605 (±0.07) | 0.5123 (±0.03) | 0.7131 (±0.24) | 0.4301 (±0.05) |
| $WEDRL_3$ | 0.7795 (±0.05) | 0.7432 (±0.03) | 0.7100 (±0.05) | 0.7100 (±0.16) | **0.4134 (±0.02)** | 0.5797 (±0.01) | 0.7976 (±0.09) | **0.4466 (±0.03)** |
| $WEDRL_5$ | 0.7540 (±0.05) | 0.7347 (±0.04) | 0.7194 (±0.05) | 0.6714 (±0.16) | 0.3954 (±0.04) | 0.5525 (±0.04) | 0.7619 (±0.10) | 0.4323 (±0.05) |
| MLP | **0.8505 (±0.06)** | **0.7822 (±0.03)** | 0.6998 (±0.03) | **0.8571 (±0.00)** | 0.2220 (±0.10) | 0.5651 (±0.14) | 0.9778 (±0.07) | 0.2358 (±0.05) |
| LSTM | 0.8167 (±0.04) | 0.7753 (±0.02) | **0.7265 (±0.02)** | **0.8571 (±0.00)** | 0.3276 (±0.09) | **0.6252 (±0.06)** | **0.9857 (±0.05)** | 0.2596 (±0.06) |

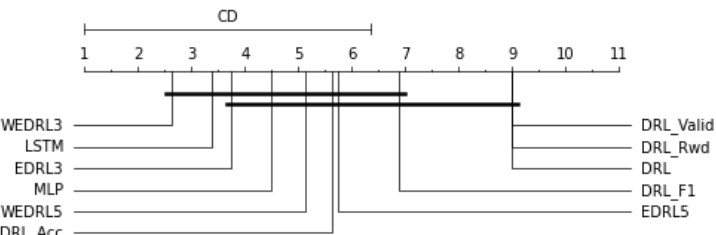

**Figure 9.** Critical difference diagram.

5.4.2. Performance on the Different Station

We then tested the generalisation ability of the best ensemble ($WEDRL_3$) with the data collected from different stations. The F1-score of each model is depicted in Table 11. We can observe that the ensemble model that was created from the model trained on CPY011 data performed well not only on their own dataset but also on CPY017, with an F1-score of 0.8200, similarly to $WEDRL_3$ on CPY012 and CPY013, which recognised anomalies on CPY014 better than their own dataset with F1-scores of 0.8421 and 0.8143, respectively. Inversely, the ensemble model on CPY014, CPY015, and CPY016 trained datasets provided poor performance when used to detect anomalies on other stations. Even though the ensemble model trained on the CPY017 dataset got an F1-score of more than 0.5 on CPY012, CPY013, and CPY014, it did not do well on many stations, with an F1-score of less than

0.3. $WEDRL_3$ scored badly not just on their own dataset but also on others, with F1-scores ranging from 0.0739 on CPY015 to 0.5748 on CPY016.

**Table 11.** The mean F1-scores of the $WEDRL_3$ models when testing with the dataset from the same station (shown in the bracket) and different stations.

| Tested Dataset | Trained Dataset | | | | | | | |
|---|---|---|---|---|---|---|---|---|
| | **CPY011** | **CPY012** | **CPY013** | **CPY014** | **CPY015** | **CPY016** | **CPY017** | **YOM009** |
| CPY011 | (0.7795) | 0.2772 | 0.2278 | 0.1718 | 0.4345 | 0.0788 | 0.2830 | 0.0900 |
| CPY012 | 0.7183 | (0.7432) | 0.6817 | 0.4624 | 0.2182 | 0.5355 | 0.6419 | 0.5096 |
| CPY013 | 0.6920 | 0.7281 | (0.7101) | 0.5512 | 0.2713 | 0.5625 | 0.5959 | 0.4816 |
| CPY014 | 0.7276 | 0.8421 | 0.8143 | (0.6962) | 0.5714 | 0.5471 | 0.7948 | 0.3450 |
| CPY015 | 0.3748 | 0.2683 | 0.1664 | 0.1482 | (0.4134) | 0.1290 | 0.1994 | 0.0739 |
| CPY016 | 0.5813 | 0.6283 | 0.6450 | 0.3327 | 0.3647 | (0.5797) | 0.4711 | 0.5748 |
| CPY017 | 0.8200 | 0.4423 | 0.4514 | 0.3803 | 0.3545 | 0.2075 | (0.7976) | 0.2931 |
| YOM009 | 0.1084 | 0.3261 | 0.3109 | 0.2619 | 0.0568 | 0.4613 | 0.3375 | (0.4466) |
| avg | **0.6002** | 0.5320 | 0.5010 | 0.3756 | 0.3356 | 0.3877 | 0.5152 | 0.3518 |
| std | 0.2426 | 0.2310 | 0.2451 | 0.1887 | 0.1551 | 0.2121 | 0.2292 | 0.1889 |

### 5.4.3. Ensemble with All Seven Models

Then, to learn more about how well the ensemble worked, we combined our developed DRL models with MLP and LSTM models. In model selection, we selected all seven models and selected the five and three models with the highest ranking in F1-score to build our ensemble model. We used the same strategy to make a final decision. So, we have 6 ensemble model for each epochs of training include majority voting ensemble model with 3 (*E3*), 5 (*E5*), and 7 (*E7*) model, and weighted ensemble model with 3 (*WE3*), 5 (*WE5*), and 7 (*WE7*) models, and the results are displayed in Table 12.

We can see that, on the CPY011 dataset, the ensemble of the top three models (E3) earned the greatest F1-score of 0.9231 with every epoch of training. On CPY012, the greatest F1-score of 0.8438 was obtained by E5 and WE7 with models trained with 10,000 epochs, and E7 with models trained with 500 epochs, while E3 and WE3 models trained with 10,000 epochs performed the best in identifying anomalies on the CPY013 dataset. With the CPY014 dataset, all ensemble models gave an F1-score of 0.8571, with the exception of the ensemble with majority voting of all seven models trained with 10,000 epochs, which performed badly with an F1-score of 0.6667. WE7 surpassed other ensemble models on the CPY015 and CPY016 datasets, with the greatest F1-score of 0.4615 and 0.6704, respectively. Every ensemble model on CPY017 produced outstanding results with a 1.0000 F1-score, particularly E3, WE3, WE5, and WE7, which produced excellent results with all training epochs. The weighted ensemble with 5 models (WE5) trained with 500 epochs performed the best on the YOM009 dataset, with a 0.5032 F1-score.

As indicated in Table 13, we averaged the F1-score of each individual model and ensemble model to compare their performance. We can observe that E3 not only performed the best model with the greatest average F1-score on all datasets but also excellently performed with a 1.0000 F1-score on CPY017 and YOM009. Among the models tested on the CPY014 dataset, the best F1-score of 0.8571 was achieved by MLP, LSTM, E3, E5, WE3, WE5, and WE7. In contrast, on the CPY015 dataset, the model with DRL-based ($DRL_{F1}$, $DRl_{Acc}$, $EDRL_3$, and $WEDRL_3$) generated the highest F1-score of 0.4134. Furthermore, as shown in Figure 10, the CD diagram was chosen to make a statistical comparison of our results, which revealed that E3 had the highest ranking, and the ensemble model that combined all seven individual models outperformed both the individual model and the ensemble model created using DRL models. It also demonstrated the ability of ensemble methods to improve the performance of individual DRL models because it represented a significant difference from individual models ($DRL$, $DRL_{Rwd}$, and $DRL_{Valid}$).

**Table 12.** The performance of the ensemble models built by combining DRL and candidate models (the best F1-score of each row is shown in bold).

| Station | Epochs | E3 | | | E5 | | | E7 | | | WE3 | | | WE5 | | | WE7 | | |
|---|---|---|---|---|---|---|---|---|---|---|---|---|---|---|---|---|---|---|---|
| | | Recall | Prec | F1 | Recall | Prec | F1 | Recall | Prec | F1 | Recall | Prec | F1 | Recall | Prec | F1 | Recall | Prec | F1 |
| CPY011 | 100 | 0.8571 | 1.0000 | **0.9231** | 0.8571 | 0.7500 | 0.8000 | 0.8571 | 0.7500 | 0.8000 | 0.8571 | 0.8571 | 0.8571 | 0.8571 | 0.8571 | 0.8571 | 0.8571 | 0.8571 | 0.8571 |
| | 500 | 0.8571 | 1.0000 | **0.9231** | 0.8571 | 0.7500 | 0.8000 | 0.8571 | 0.7500 | 0.8000 | 0.8571 | 0.8571 | 0.8571 | 0.8571 | 0.8571 | 0.8571 | 0.8571 | 0.8571 | 0.8571 |
| | 1000 | 0.8571 | 1.0000 | **0.9231** | 0.8571 | 1.0000 | **0.9231** | 0.8571 | 1.0000 | **0.9231** | 0.8571 | 0.8571 | 0.8571 | 0.8571 | 0.8571 | 0.8571 | 0.8571 | 0.8571 | 0.8571 |
| | 5000 | 0.8571 | 1.0000 | **0.9231** | 0.7143 | 0.7143 | 0.7143 | 0.8571 | 0.7500 | 0.8000 | 0.8571 | 0.8571 | 0.8571 | 0.8571 | 0.8571 | 0.8571 | 0.8571 | 0.8571 | 0.8571 |
| | 10,000 | 0.8571 | 1.0000 | **0.9231** | 0.8571 | 0.8571 | 0.8571 | 0.8571 | 0.8571 | 0.8571 | 0.8571 | 0.8571 | 0.8571 | 0.8571 | 0.8571 | 0.8571 | 0.8571 | 0.8571 | 0.8571 |
| | avg | 0.8571 | 1.0000 | **0.9231** | 0.8285 | 0.8143 | 0.8189 | 0.8571 | 0.8214 | 0.8360 | 0.8571 | 0.8571 | 0.8571 | 0.8571 | 0.8571 | 0.8571 | 0.8571 | 0.8571 | 0.8571 |
| | std | 0.0000 | 0.0000 | 0.0000 | 0.0639 | 0.1168 | 0.0774 | 0.0000 | 0.1101 | 0.0546 | 0.0000 | 0.0000 | 0.0000 | 0.0000 | 0.0000 | 0.0000 | 0.0000 | 0.0000 | 0.0000 |
| CPY012 | 100 | 0.7647 | 0.9286 | **0.8387** | 0.7647 | 0.8387 | 0.8000 | 0.7941 | 0.7714 | 0.7826 | 0.7647 | 0.8966 | 0.8254 | 0.7647 | 0.8966 | 0.8254 | 0.7647 | 0.8966 | 0.8254 |
| | 500 | 0.7647 | 0.9286 | 0.8387 | 0.7941 | 0.7297 | 0.7606 | 0.7941 | 0.9000 | **0.8438** | 0.7647 | 0.8966 | 0.8254 | 0.7647 | 0.8966 | 0.8254 | 0.7647 | 0.8966 | 0.8254 |
| | 1000 | 0.7647 | 0.9286 | **0.8387** | 0.7647 | 0.8966 | 0.8254 | 0.7647 | 0.8966 | 0.8254 | 0.7647 | 0.8966 | 0.8254 | 0.7647 | 0.8966 | 0.8254 | 0.7647 | 0.8966 | 0.8254 |
| | 5000 | 0.7647 | 0.9286 | **0.8387** | 0.7059 | 0.7500 | 0.7273 | 0.7353 | 0.8333 | 0.7812 | 0.7647 | 0.8966 | 0.8254 | 0.7647 | 0.8966 | 0.8254 | 0.7647 | 0.8966 | 0.8254 |
| | 10,000 | 0.7647 | 0.9286 | 0.8387 | 0.7941 | 0.9000 | **0.8438** | 0.7059 | 0.8276 | 0.7619 | 0.7647 | 0.8966 | 0.8254 | 0.7647 | 0.8966 | 0.8254 | 0.7941 | 0.9000 | **0.8438** |
| | avg | 0.7647 | 0.9286 | **0.8387** | 0.7647 | 0.8230 | 0.7914 | 0.7588 | 0.8458 | 0.7990 | 0.7647 | 0.8966 | 0.8254 | 0.7647 | 0.8966 | 0.8254 | 0.7706 | 0.8973 | 0.8291 |
| | std | 0.0000 | 0.0000 | 0.0000 | 0.0360 | 0.0800 | 0.0475 | 0.0383 | 0.0537 | 0.0342 | 0.0000 | 0.0000 | 0.0000 | 0.0000 | 0.0000 | 0.0000 | 0.0131 | 0.0015 | 0.0082 |
| CPY013 | 100 | 0.7742 | 0.7273 | 0.7500 | 0.8387 | 0.6842 | **0.7536** | 0.8065 | 0.6757 | 0.7353 | 0.7419 | 0.6765 | 0.7077 | 0.7742 | 0.6857 | 0.7273 | 0.7742 | 0.6857 | 0.7273 |
| | 500 | 0.7742 | 0.7273 | **0.7500** | 0.8387 | 0.6341 | 0.7222 | 0.7419 | 0.6970 | 0.7188 | 0.7419 | 0.6765 | 0.7077 | 0.7419 | 0.6765 | 0.7077 | 0.7419 | 0.6765 | 0.7077 |
| | 1000 | 0.7742 | 0.7273 | 0.7500 | 0.8710 | 0.6750 | **0.7606** | 0.8065 | 0.6410 | 0.7143 | 0.7419 | 0.6765 | 0.7077 | 0.7742 | 0.6857 | 0.7273 | 0.7742 | 0.6857 | 0.7273 |
| | 5000 | 0.7742 | 0.7273 | **0.7500** | 0.7419 | 0.6765 | 0.7077 | 0.7419 | 0.6765 | 0.7077 | 0.7419 | 0.6765 | 0.7077 | 0.7419 | 0.6765 | 0.7077 | 0.7742 | 0.6857 | 0.7273 |
| | 10,000 | 0.8387 | 0.7647 | **0.8000** | 0.7742 | 0.7742 | 0.7742 | 0.7742 | 0.7742 | 0.7742 | 0.8387 | 0.7647 | **0.8000** | 0.7742 | 0.7273 | 0.7500 | 0.7419 | 0.6765 | 0.7077 |
| | avg | 0.7871 | 0.7348 | **0.7600** | 0.8129 | 0.6888 | 0.7437 | 0.7742 | 0.6929 | 0.7301 | 0.7613 | 0.6941 | 0.7262 | 0.7613 | 0.6903 | 0.7240 | 0.7613 | 0.6820 | 0.7195 |
| | std | 0.0288 | 0.0167 | 0.0224 | 0.0530 | 0.0516 | 0.0277 | 0.0323 | 0.0424 | 0.0277 | 0.0177 | 0.0212 | 0.0175 | 0.0177 | 0.0050 | 0.0107 | 0.0177 | 0.0050 | 0.0107 |
| CPY014 | 100 | 0.7500 | 1.0000 | **0.8571** | 0.7500 | 1.0000 | **0.8571** | 0.7500 | 1.0000 | **0.8571** | 0.7500 | 1.0000 | **0.8571** | 0.7500 | 1.0000 | **0.8571** | 0.7500 | 1.0000 | **0.8571** |
| | 500 | 0.7500 | 1.0000 | **0.8571** | 0.7500 | 1.0000 | **0.8571** | 0.7500 | 1.0000 | **0.8571** | 0.7500 | 1.0000 | **0.8571** | 0.7500 | 1.0000 | **0.8571** | 0.7500 | 1.0000 | **0.8571** |
| | 1000 | 0.7500 | 1.0000 | **0.8571** | 0.7500 | 1.0000 | **0.8571** | 0.7500 | 1.0000 | **0.8571** | 0.7500 | 1.0000 | **0.8571** | 0.7500 | 1.0000 | **0.8571** | 0.7500 | 1.0000 | **0.8571** |
| | 5000 | 0.7500 | 1.0000 | **0.8571** | 0.7500 | 1.0000 | **0.8571** | 0.7500 | 1.0000 | **0.8571** | 0.7500 | 1.0000 | **0.8571** | 0.7500 | 1.0000 | **0.8571** | 0.7500 | 1.0000 | **0.8571** |
| | 10,000 | 0.7500 | 1.0000 | **0.8571** | 0.7500 | 1.0000 | **0.8571** | 0.7500 | 0.6000 | 0.6667 | 0.7500 | 1.0000 | **0.8571** | 0.7500 | 1.0000 | **0.8571** | 0.7500 | 1.0000 | **0.8571** |
| | avg | 0.7500 | 1.0000 | **0.8571** | 0.7500 | 1.0000 | **0.8571** | 0.7500 | 0.9200 | 0.8190 | 0.7500 | 1.0000 | **0.8571** | 0.7500 | 1.0000 | **0.8571** | 0.7500 | 1.0000 | **0.8571** |
| | std | 0.0000 | 0.0000 | 0.0000 | 0.0000 | 0.0000 | 0.0000 | 0.0000 | 0.1789 | 0.0851 | 0.0000 | 0.0000 | 0.0000 | 0.0000 | 0.0000 | 0.0000 | 0.0000 | 0.0000 | 0.0000 |
| CPY015 | 100 | 0.3235 | 0.5238 | **0.4000** | 0.2647 | 0.5000 | 0.3462 | 0.2647 | 0.5000 | 0.3462 | 0.3235 | 0.5238 | **0.4000** | 0.2647 | 0.4737 | 0.3396 | 0.2353 | 0.4444 | 0.3077 |
| | 500 | 0.3235 | 0.5000 | **0.3929** | 0.2353 | 0.5000 | 0.3200 | 0.2353 | 0.5000 | 0.3200 | 0.3235 | 0.5000 | **0.3929** | 0.2941 | 0.5263 | 0.3774 | 0.2647 | 0.5000 | 0.3462 |
| | 1000 | 0.3824 | 0.5000 | 0.4333 | 0.4118 | 0.4828 | **0.4444** | 0.2647 | 0.4286 | 0.3273 | 0.3824 | 0.5000 | 0.4333 | 0.3824 | 0.4643 | 0.4194 | 0.3824 | 0.4333 | 0.4062 |
| | 5000 | 0.3235 | 0.5238 | **0.4000** | 0.2647 | 0.5000 | 0.3462 | 0.2647 | 0.5000 | 0.3462 | 0.3235 | 0.5238 | **0.4000** | 0.2941 | 0.5000 | 0.3704 | 0.2941 | 0.5000 | 0.3704 |
| | 10,000 | 0.3824 | 0.5200 | 0.4407 | 0.3824 | 0.4483 | 0.4127 | 0.3529 | 0.4800 | 0.4068 | 0.3824 | 0.5200 | 0.4407 | 0.3824 | 0.5200 | 0.4407 | 0.4412 | 0.4839 | **0.4615** |
| | avg | 0.3471 | 0.5135 | **0.4134** | 0.3118 | 0.4862 | 0.3739 | 0.2765 | 0.4817 | 0.3493 | 0.3471 | 0.5135 | **0.4134** | 0.3235 | 0.4969 | 0.3895 | 0.3235 | 0.4723 | 0.3784 |
| | std | 0.0323 | 0.0124 | 0.0219 | 0.0795 | 0.0225 | 0.0522 | 0.0446 | 0.0309 | 0.0342 | 0.0323 | 0.0124 | 0.0219 | 0.0551 | 0.0274 | 0.0404 | 0.0858 | 0.0315 | 0.0587 |
| CPY016 | 100 | 0.5421 | 0.8529 | **0.6629** | 0.5981 | 0.5470 | 0.5714 | 0.6075 | 0.5462 | 0.5752 | 0.5421 | 0.8406 | 0.6591 | 0.5421 | 0.8286 | 0.6554 | 0.5421 | 0.8286 | 0.6554 |
| | 500 | 0.5421 | 0.8529 | **0.6629** | 0.5981 | 0.6154 | 0.6066 | 0.5888 | 0.6632 | 0.6238 | 0.5421 | 0.8406 | 0.6591 | 0.5421 | 0.8406 | 0.6591 | 0.5421 | 0.8406 | 0.6591 |
| | 1000 | 0.5421 | 0.8529 | **0.6629** | 0.5794 | 0.5794 | 0.5794 | 0.6075 | 0.5372 | 0.5702 | 0.5421 | 0.8406 | 0.6591 | 0.5421 | 0.8286 | 0.6554 | 0.5421 | 0.8286 | 0.6554 |
| | 5000 | 0.5421 | 0.8529 | 0.6629 | 0.5607 | 0.7059 | 0.6250 | 0.5607 | 0.6122 | 0.5854 | 0.5421 | 0.8406 | 0.6591 | 0.5421 | 0.8286 | 0.6554 | 0.5607 | 0.8333 | **0.6704** |
| | 10,000 | 0.5421 | 0.8529 | **0.6629** | 0.6075 | 0.5752 | 0.5909 | 0.5981 | 0.4638 | 0.5224 | 0.5421 | 0.8406 | 0.6591 | 0.5421 | 0.8286 | 0.6554 | 0.5421 | 0.8529 | **0.6629** |
| | avg | 0.5421 | 0.8529 | **0.6629** | 0.5888 | 0.6046 | 0.5947 | 0.5925 | 0.5645 | 0.5754 | 0.5421 | 0.8406 | 0.6591 | 0.5421 | 0.8310 | 0.6561 | 0.5458 | 0.8368 | 0.6606 |
| | std | 0.0000 | 0.0000 | 0.0000 | 0.0187 | 0.0616 | 0.0215 | 0.0194 | 0.0762 | 0.0215 | 0.0000 | 0.0000 | 0.0000 | 0.0000 | 0.0054 | 0.0017 | 0.0083 | 0.0103 | 0.0063 |
| CPY017 | 100 | 1.0000 | 1.0000 | **1.0000** | 1.0000 | 1.0000 | **1.0000** | 1.0000 | 0.7500 | 0.8571 | 1.0000 | 1.0000 | **1.0000** | 1.0000 | 1.0000 | **1.0000** | 1.0000 | 1.0000 | **1.0000** |
| | 500 | 1.0000 | 1.0000 | **1.0000** | 1.0000 | 1.0000 | **1.0000** | 1.0000 | 1.0000 | **1.0000** | 1.0000 | 1.0000 | **1.0000** | 1.0000 | 1.0000 | **1.0000** | 1.0000 | 1.0000 | **1.0000** |
| | 1000 | 1.0000 | 1.0000 | **1.0000** | 1.0000 | 0.7500 | 0.8571 | 1.0000 | 0.7500 | 0.8571 | 1.0000 | 1.0000 | **1.0000** | 1.0000 | 1.0000 | **1.0000** | 1.0000 | 1.0000 | **1.0000** |
| | 5000 | 1.0000 | 1.0000 | **1.0000** | 1.0000 | 0.7500 | 0.8571 | 1.0000 | 0.7500 | 0.8571 | 1.0000 | 1.0000 | **1.0000** | 1.0000 | 1.0000 | **1.0000** | 1.0000 | 1.0000 | **1.0000** |
| | 10,000 | 1.0000 | 1.0000 | **1.0000** | 1.0000 | 1.0000 | **1.0000** | 0.6667 | 1.0000 | 0.8000 | 1.0000 | 1.0000 | **1.0000** | 1.0000 | 1.0000 | **1.0000** | 1.0000 | 1.0000 | **1.0000** |
| | avg | 1.0000 | 1.0000 | **1.0000** | 1.0000 | 0.9000 | 0.9428 | 0.9333 | 0.8500 | 0.8743 | 1.0000 | 1.0000 | **1.0000** | 1.0000 | 1.0000 | **1.0000** | 1.0000 | 1.0000 | **1.0000** |
| | std | 0.0000 | 0.0000 | 0.0000 | 0.0000 | 0.1369 | 0.0783 | 0.1491 | 0.1369 | 0.0745 | 0.0000 | 0.0000 | 0.0000 | 0.0000 | 0.0000 | 0.0000 | 0.0000 | 0.0000 | 0.0000 |
| YOM009 | 100 | 0.6308 | 0.3254 | 0.4293 | 0.5846 | 0.3115 | 0.4064 | 0.5538 | 0.3303 | 0.4138 | 0.6154 | 0.3226 | 0.4233 | 0.6462 | 0.3281 | **0.4352** | 0.5846 | 0.3065 | 0.4021 |
| | 500 | 0.4769 | 0.4769 | 0.4769 | 0.6154 | 0.3960 | 0.4819 | 0.3538 | 0.4694 | 0.4035 | 0.4769 | 0.4769 | 0.4769 | 0.6000 | 0.4333 | **0.5032** | 0.6154 | 0.4082 | 0.4908 |
| | 1000 | 0.5538 | 0.3600 | 0.4364 | 0.5538 | 0.3186 | 0.4045 | 0.4769 | 0.3875 | 0.4276 | 0.5231 | 0.3778 | **0.4387** | 0.4923 | 0.3299 | 0.3951 | 0.5538 | 0.3396 | 0.4211 |
| | 5000 | 0.5538 | 0.4091 | 0.4706 | 0.4923 | 0.4384 | 0.4638 | 0.4462 | 0.4531 | 0.4496 | 0.6308 | 0.3981 | **0.4881** | 0.5538 | 0.3789 | 0.4500 | 0.5385 | 0.4430 | 0.4861 |
| | 10,000 | 0.5385 | 0.2991 | 0.3846 | 0.4923 | 0.3048 | 0.3765 | 0.4154 | 0.3293 | 0.3673 | 0.4154 | 0.3971 | **0.4060** | 0.4615 | 0.3158 | 0.3750 | 0.5692 | 0.3186 | 0.4045 |
| | avg | 0.5508 | 0.3741 | 0.4396 | 0.5477 | 0.3539 | 0.4266 | 0.4492 | 0.3939 | 0.4124 | 0.5323 | 0.3945 | **0.4466** | 0.5508 | 0.3572 | 0.4317 | 0.5692 | 0.3632 | 0.4409 |
| | std | 0.0548 | 0.0707 | 0.0371 | 0.0550 | 0.0599 | 0.0443 | 0.0741 | 0.0661 | 0.0305 | 0.0914 | 0.0554 | 0.0350 | 0.0757 | 0.0489 | 0.0500 | 0.0308 | 0.0595 | 0.0440 |

**Table 13.** The mean F1-scores and standard deviation of all models when testing with the dataset from different stations (the best F1-score of each station is shown in bold).

| Models | CPY011 | CPY012 | CPY013 | CPY014 | CPY015 | CPY016 | CPY017 | YOM009 |
|---|---|---|---|---|---|---|---|---|
| $DRL$ | 0.7433 (±0.08) | 0.6550 (±0.08) | 0.6014 (±0.10) | 0.4823 (±0.20) | 0.3477 (±0.06) | 0.3770 (±0.03) | 0.6801 (±0.21) | 0.3581 (±0.06) |
| $DRL_{F1}$ | 0.7170 (±0.07) | 0.7146 (±0.06) | 0.6963 (±0.07) | 0.6733 (±0.08) | **0.4134 (±0.02)** | 0.5438 (±0.05) | 0.6381 (±0.16) | 0.4152 (±0.04) |
| $DRLRwd$ | 0.7433 (±0.08) | 0.6468 (±0.10) | 0.6212 (±0.09) | 0.5537 (±0.26) | 0.3477 (±0.06) | 0.3499 (±0.06) | 0.6762 (±0.21) | 0.3891 (±0.04) |
| $DRL_{Acc}$ | 0.7170 (±0.07) | 0.7234 (±0.04) | 0.6920 (±0.07) | 0.6733 (±0.08) | **0.4134 (±0.02)** | 0.5714 (±0.02) | 0.7219 (±0.14) | 0.4248 (±0.04) |
| $DRL_{Valid}$ | 0.6020 (±0.17) | 0.7045 (±0.10) | 0.5639 (±0.21) | 0.6596 (±0.14) | 0.3211 (±0.10) | 0.4867 (±0.07) | 0.6488 (±0.15) | 0.4378 (±0.03) |
| MLP | 0.8505 (±0.06) | 0.7822 (±0.03) | 0.6998 (±0.03) | **0.8571 (±0.00)** | 0.2220 (±0.05) | 0.5651 (±0.14) | 0.9778 (±0.07) | 0.2358 (±0.05) |
| LSTM | 0.8167 (±0.04) | 0.7753 (±0.02) | 0.7265 (±0.02) | **0.8571 (±0.00)** | 0.3276 (±0.09) | 0.6252 (±0.06) | 0.9857 (±0.05) | 0.2596 (±0.06) |
| $EDRL_3$ | 0.7700 (±0.06) | 0.7367 (±0.02) | 0.7033 (±0.07) | 0.6948 (±0.11) | **0.4134 (±0.02)** | 0.5544 (±0.03) | 0.7809 (±0.10) | 0.4396 (±0.04) |
| $EDRL_5$ | 0.7455 (±0.09) | 0.7558 (±0.03) | 0.6819 (±0.03) | 0.7295 (±0.12) | 0.3605 (±0.07) | 0.5123 (±0.03) | 0.7131 (±0.24) | 0.4301 (±0.05) |
| E3 | **0.9231 (±0.00)** | **0.8387 (±0.00)** | **0.7600 (±0.02)** | **0.8571 (±0.00)** | **0.4134 (±0.02)** | **0.6629 (±0.00)** | **1.0000 (±0.00)** | **1.0000 (±0.04)** |
| E5 | 0.8189 (±0.08) | 0.7914 (±0.05) | 0.7437 (±0.03) | **0.8571 (±0.00)** | 0.3739 (±0.05) | 0.5947 (±0.02) | 0.9428 (±0.08) | 0.9428 (±0.04) |
| E7 | 0.8360 (±0.05) | 0.7990 (±0.03) | 0.7301 (±0.03) | 0.8190 (±0.09) | 0.3493 (±0.03) | 0.5754 (±0.04) | 0.8743 (±0.07) | 0.8743 (±0.03) |
| $WEDRL_3$ | 0.7795 (±0.05) | 0.7432 (±0.03) | 0.7100 (±0.07) | 0.6962 (±0.16) | **0.4134 (±0.02)** | 0.5797 (±0.01) | 0.7976 (±0.09) | 0.4466 (±0.03) |
| $WEDRL_5$ | 0.7540 (±0.05) | 0.7347 (±0.04) | 0.7194 (±0.05) | 0.6714 (±0.16) | 0.3954 (±0.04) | 0.5525 (±0.04) | 0.7619 (±0.10) | 0.4323 (±0.05) |
| WE3 | 0.8571 (±0.00) | 0.8254 (±0.00) | 0.7262 (±0.04) | **0.8571 (±0.00)** | **0.4134 (±0.02)** | 0.6591 (±0.00) | **1.0000 (±0.00)** | **1.0000 (±0.03)** |
| WE5 | 0.8571 (±0.00) | 0.8254 (±0.00) | 0.7240 (±0.02) | **0.8571 (±0.00)** | 0.3895 (±0.04) | 0.6561 (±0.00) | **1.0000 (±0.00)** | **1.0000 (±0.05)** |
| WE7 | 0.8571 (±0.00) | 0.8291 (±0.01) | 0.7195 (±0.01) | **0.8571 (±0.00)** | 0.3784 (±0.06) | 0.6606 (±0.01) | **1.0000 (±0.00)** | **1.0000 (±0.04)** |

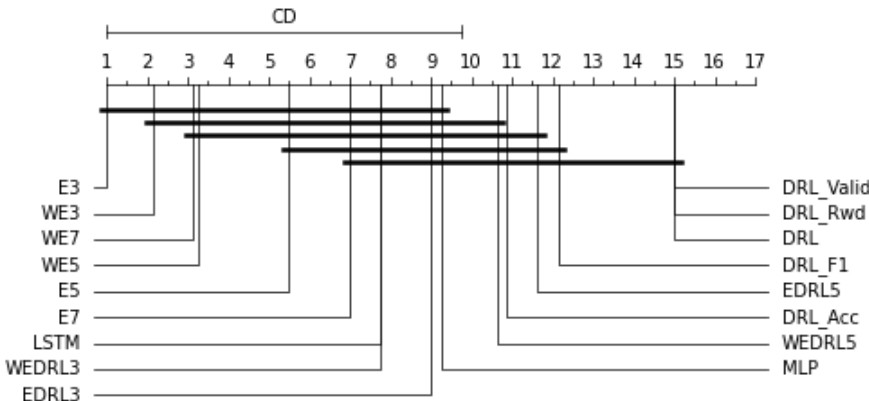

**Figure 10.** A critical difference diagram.

We then tested the generalisation ability of ensemble models with the data collected from different stations. The F1-score of each station is depicted in Table 14. Using ensemble E3 with CPY011 data, to identify anomalies from other stations, we can see that it works well with F1-scores of more than 0.5800, but it performed poorly at detecting anomalies on CPY015 and YOM009 with F1-scores of 0.3444 and 0.1017, respectively. E3 on CPY012 performed well when detecting anomalies on CPY014 with a 0.8635 F1-score. Similarly, E3 on CPY013 provided a higher F1-score on their own dataset when detecting anomalies on CPY012 and CPY014 with an F1-score of 0.8060 and 0.8571, respectively. The best ensemble on CPY014 generated excellent performance when identifying anomalies on CPY017 data. In contrast, E3 on CPY015 performed poorly on YOM009 with an F1-score of only 0.0437. While considered E3 on CPY016, although it provided good performance with an F1-score higher than 0.6 on CPY012, CPY013, and CPY014, it performed poorly on CPY011, CPY015, CPY017, and YOM009 with an F1-score lower than 0.45. E3 on CPY017 provided good results with an F1-score of more than 0.69, except on CPY015, CPY016, and YOM009 with an F1-score lower than 0.56. Meanwhile, E3 on YOM009 generated an F1-score on its own of only 0.43, but it performed excellently when detecting anomalies on CPY017 and other datasets with an F1-score higher than 0.65, except on CPY015 with an F1-score of 0.2414.

**Table 14.** The F1-scores of the E3 models when testing with the dataset from the same station (shown in the bracket) and different stations.

| Tested Dataset | Trained Dataset | | | | | | | |
|---|---|---|---|---|---|---|---|---|
| | CPY011 | CPY012 | CPY013 | CPY014 | CPY015 | CPY016 | CPY017 | YOM009 |
| CPY011 | (0.9231) | 0.4000 | 0.3222 | 0.7778 | 0.4028 | 0.1373 | 0.7059 | 0.6087 |
| CPY012 | 0.7183 | (0.8387) | 0.8060 | 0.7500 | 0.1508 | 0.6923 | 0.7273 | 0.7857 |
| CPY013 | 0.6920 | 0.7350 | (0.7600) | 0.7000 | 0.2117 | 0.7123 | 0.6909 | 0.7719 |
| CPY014 | 0.6895 | 0.8635 | 0.8571 | (0.8571) | 0.5714 | 0.6000 | 0.8571 | 0.8571 |
| CPY015 | 0.3444 | 0.2192 | 0.2093 | 0.2593 | (0.4132) | 0.1420 | 0.3077 | 0.2414 |
| CPY016 | 0.5840 | 0.6398 | 0.6603 | 0.6322 | 0.3495 | (0.6629) | 0.5665 | 0.6580 |
| CPY017 | 0.7600 | 0.6667 | 0.6000 | 1.0000 | 0.3651 | 0.2857 | (1.0000) | 1.0000 |
| YOM009 | 0.1017 | 0.2948 | 0.3221 | 0.2908 | 0.0437 | 0.4500 | 0.3230 | (0.4396) |
| avg | 0.6016 | 0.5822 | 0.5671 | 0.6584 | 0.3135 | 0.4603 | 0.6473 | **0.6703** |
| std | 0.2603 | 0.2468 | 0.2496 | 0.2607 | 0.1682 | 0.2432 | 0.2412 | 0.2415 |

## 6. Discussion

We can observe that when the number of training epochs increases, the performance of each model grows or decreases in each epoch, then drops and bounces back. This might indicate that our model is still learning or is learning too much—that is, it is difficult to decide when it is time to stop training.

Even though DRL can do better than other models, it is time-consuming—at least 50 times slower than MLP models on average—because we have to train it until it performs well enough and we cannot predict how long that will take. The size of the windows must also be taken into account. A larger window size takes more time than a smaller window size. The window size has an effect on the comparison of data in windows to identify the anomaly. Additionally, we may add additional neural networks to improve the accuracy of our technique, but training will take longer.

DRL does better than other models when it is trained on datasets with a low number of outliers. This proves the ability to detect unknown anomalies. However, its performance is insufficient, which may be due to an imbalance in our dataset. As a result, models may lack sufficient information to explore and leverage knowledge for adaptive detection of unknown abnormalities.

Moreover, the neural structure that works well with one station may not function well with another. Hence, the problems of this topic include determining the suitable neural structure for each station. Furthermore, the primary parameter that requires further attention is the reward function, since a suitable reward will impact the model's learning process.

In the case of ensemble models, when all of the individual models in an ensemble perform similarly, majority voting is the best method for determining the final decision. However, when the accuracies of individual models are different, the weighted voting is the best way to utilise the strengths of the good models in making a decision. Furthermore, the ensemble model can also reduce the false alarm rate, as seen by an increased precision score. It should be noted that, although single models performed well on certain stations, they did poorly on others, such as the LSTM model. As a result, we cannot rely on a single model since we do not know if it is the best or not. The ensemble models, on the other hand, are more reliable, even though they may not produce the best accuracy for every station. On the whole, nevertheless, most ensembles, such as $WEDRL_3$ performed consistently very well and their accuracies are always ranked highly at every station, whilst the individual models: DRL, MLP and LSTM, are not consistent through out all the stations.

## 7. Conclusions

In this research, we firstly investigated how deep reinforcement learning (DRL) can be applied to detect anomalies in water level data and then devised two strategies to construct more effective and reliable ensembles. For DRL, we defined a reward function as it plays a key role in determining the success of an RL. We developed ensemble models with five deep reinforcement learning models, generated by the same DRL algorithm but with different criteria of performance measurement. We tested our ensemble approach on telemetry water level data from eight different stations. We compared our approach to two different neural network models. Moreover, we demonstrate the ability to detect unknown anomalies by using the trained model to detect anomalies from other stations' data.

The results indicate that $DRL_{Acc}$ models are the best individual DRL models, but they performed slightly poor than LSTM. When tested on different stations, LSTM still does better than others, but its accuracy is not satisfactory. When compared to an ensemble approach, LSTM was more accurate in some stations than other ensembles with DRL models, but less accurate in some others. On the whole, the statistical results from the CD diagram showed that our ensemble approach with only 3 members of DRL models, $WEDRL_3$, was superior. Furthermore, all ensemble models that were combined by selecting models from 5 DRL models, MLP, and LSTM outperformed both the best individual model, LSTM, and the best ensemble using DRL models, $WEDRL_3$. This is supported by the highest F1-score and rankings with the CD diagram. It is clear that ensemble methods not only increased the accuracy of a single model but also provided a higher reliability of performance.

In conclusion, DRL is applicable for detecting anomalies in telemetry water level data with added benefit of detecting unknown anomalies. Our ensemble construction methods

can be used to build ensemble models from selected single DRL models in order to increase the accuracy and reliability. In general, the ensembles are consistent in producing more accurate classification, although they may not always achieve the best results. Moreover, they are superior in reducing the number of false alarms in identifying abnormalities in water level data, which is very important in real application. The next stage in our study will be to develop more effective and efficient techniques for correcting the identified anomalies in the data.

**Author Contributions:** Conceptualization, T.K. and W.W.; methodology, T.K. and W.W.; formal analysis, T.K. and W.W.; investigation, T.K. and W.W.; resources, T.K.; writing and revision: T.K. and final revision: W.W.; project administration, T.K. All authors have read and agreed to the published version of the manuscript.

**Funding:** This research received no external funding.

**Institutional Review Board Statement:** Not applicable.

**Informed Consent Statement:** Not applicable.

**Data Availability Statement:** Data available on request due to restriction, e.g., privacy. The data presented in this study are available on request from the Hydro-Informatics Institute (HII).

**Acknowledgments:** The authors would like to thank the Hydro-Informatics Institute of Ministry of Higher Education, Science, Research and Innovation, Thailand, for providing the scholarship for Thakolpat Khampuengson to do his Ph.D. at the university of East Anglia.

**Conflicts of Interest:** The authors declare no conflict of interest.

## Abbreviations

The following abbreviations are used in this manuscript:

| | |
|---|---|
| CNN | Convolutional Neural Network |
| BDRL | Best Deep Reinforcement Learning model |
| DQN | Deep Q-Learning Network |
| DRL | Deep Reinforcement Learning |
| HII | Hydro Informatics Institute |
| LSTM | Long-Short Term Memory |
| MLP | Multilayer Perceptron |
| NN | Neural Network |
| RL | Reinforcement Learning |

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
