# Peer review of "Deep Reinforcement Learning Ensemble for Detecting Anomaly in Telemetry Water Level Data"

_water, doi:10.3390/w14162492_

Round 1
Reviewer 1 Report
The authors are to be complimented on producing an interesting paper relevant to the readership of Water.
Author Response
We appreciate the time and effort that you have dedicated to providing your valuable feedback on my manuscript.
Reviewer 2 Report
This study can be published if the authors will publically make some of the data. An additional demonstration section showing the analysis on a small publically domain dataset, so that results can be reproduced, is needed. Without that, I am against publishing this work.
Author Response
We appreciate the time and effort that you have dedicated to providing your valuable feedback on my manuscript.
Here is a point-by-point response to the your comments and concerns.
Point 1: This study can be published if the authors will publically make some of the data. An additional demonstration section showing the analysis on a small publically domain dataset, so that results can be reproduced, is needed. Without that, I am against publishing this work.
Response 1: We understand and appreciate your constructive comments/suggestions, particularly on adding a demonstration case with another small dataset from public domains. We then tried to search through the internet, but unfortunately we could not find any suitable data for this purpose.
We also tried to look some studies similar to us and found that they all used their own data and have not made their data available for other researchers.
So, it looks very difficult in a short time to do that, although we will keep looking around for similar data to perform that demonstration.
That’s one of the reasons we are happy to accept your suggestions to make some of our data public so that they can be used by other interested researchers.

Reviewer 3 Report
The paper proposes deep reinforcement learning models for anomaly detection in telemetry water levels. The proposed models are compared with two baseline models, i.e., MLP and LSTM.
In the introduction, please add the research questions you are trying to answer and the main objectives of this work.
The manuscript misses a "Related work" section. Although some related works are discussed in the Introduction section, I advise the authors to add such a section.
The current literature proposes to detect change points in the time series to minimize the number of false alarms. I find no mention of such works in the current manuscript. Thus, I recommend the authors to review the current literature and improve the related work by discussing how anomaly detection is enhanced when addressing the issues raised by change points. Please see and reference some of the articles found here [1]
There is no mention of variational autoencoders. I advise the authors to also discuss such models.
I advise the authors to improve the related work by adding more references.
[1] https://scholar.google.com/scholar?hl=en&as_sdt=0%2C5&q=Anomaly+Detection+%2B+Change+Point++detection+%2B+Time+Series&btnG=
Author Response
We appreciate the time and effort that you have dedicated to providing your valuable feedback on my manuscript.
Please see the attachment for point-by-point response to the your comments and concerns.

Round 2
Reviewer 2 Report
The paper can now be accepted.
Reviewer 3 Report
I congratulate the authors for responding to all my comments and improving their manuscript accordingly to my observations.
I have no further comments and I believe the manuscript can be published in its current form.